# MTL neurons phase-lock to human hippocampal theta

Daniel R Schonhaut[1]*, Aditya M Rao[2], Ashwin G Ramayya[3], Ethan A Solomon[4], Nora A Herweg[2], Itzhak Fried[5,6], Michael J Kahana[2]*

[1]Department of Neuroscience, Perelman School of Medicine, University of Pennsylvania, Philadelphia, United States; [2]Department of Psychology, University of Pennsylvania, Philadelphia, United States; [3]Department of Neurosurgery, University of Pennsylvania, Philadelphia, United States; [4]Department of Bioengineering, University of Pennsylvania, Philadelphia, United States; [5]Department of Neurosurgery, Neurosurgery, David Geffen School of Medicine and Semel Institute for Neuroscience and Human Behavior, University of California, Los Angeles, Los Angeles, United States; [6]Faculty of Medicine, Tel-Aviv University, Tel-Aviv, Israel

**\*For correspondence:**
daniel.schonhaut@gmail.com
(DRS);
kahana@psych.upenn.edu (MJK)

**Competing interest:** The authors declare that no competing interests exist.

**Abstract** Memory formation depends on neural activity across a network of regions, including the hippocampus and broader medial temporal lobe (MTL). Interactions between these regions have been studied indirectly using functional MRI, but the bases for interregional communication at a cellular level remain poorly understood. Here, we evaluate the hypothesis that oscillatory currents in the hippocampus synchronize the firing of neurons both within and outside the hippocampus. We recorded extracellular spikes from 1854 single- and multi-units simultaneously with hippocampal local field potentials (LFPs) in 28 neurosurgical patients who completed virtual navigation experiments. A majority of hippocampal neurons phase-locked to oscillations in the slow (2–4 Hz) or fast (6–10 Hz) theta bands, with a significant subset exhibiting nested slow theta × beta frequency (13–20 Hz) phase-locking. Outside of the hippocampus, phase-locking to hippocampal oscillations occurred only at theta frequencies and primarily among neurons in the entorhinal cortex and amygdala. Moreover, extrahippocampal neurons phase-locked to hippocampal theta even when theta did not appear locally. These results indicate that spike-time synchronization with hippocampal theta is a defining feature of neuronal activity in the hippocampus and structurally connected MTL regions. Theta phase-locking could mediate flexible communication with the hippocampus to influence the content and quality of memories.

## Editor's evaluation

Large sample size electrophysiology in human brains is rare, and thus this work is an important contribution to the field. The mesoscopic descriptive analyses provide a convincing bridge to work done in other species and will likely further contribute to its long term value to the field.

## Introduction

The hippocampus is the operational hub of a spatially distributed episodic memory system that enables us to remember past experiences in rich detail, together with the place and time at which they occurred (*Eichenbaum, 2000*; *Moscovitch et al., 2016*). To serve in this capacity, the hippocampus must maintain precise but flexible connections with the rest of the memory system. Understanding the mechanisms that govern connections among regions supporting episodic memory is a major concern

of systems neuroscience, and could accelerate efforts to develop treatments for memory disorders and age-related memory decline.

A leading hypothesis is that theta (2–10 Hz) oscillations within the hippocampus facilitate interactions between the hippocampus and other brain regions (*Buzsáki, 2010*; *Fell and Axmacher, 2011*; *Moscovitch et al., 2016*). Hippocampal neurons are more receptive to synaptic excitation at specific theta phases (*Kamondi et al., 1998*), so well-timed inputs can more effectively drive activity than inputs at random phases (*Fries, 2005*). Long-term potentiation and long-term depression in the rodent hippocampus also depend on theta phase (*Pavlides et al., 1988*; *Huerta and Lisman, 1995*; *Hyman et al., 2003*), offering a possible link between the phase at which inputs arrive and the strength of their encoding. Experimental evidence for this hypothesis comes largely from studies in the rat medial prefrontal cortex (mPFC), a downstream target of hippocampal area CA1. mPFC neurons phase-lock to hippocampal theta during short-term memory tasks (*Siapas et al., 2005*; *Hyman et al., 2005*; *Sirota et al., 2008*), and stronger phase-locking predicts better performance (*Jones and Wilson, 2005*; *Hyman et al., 2010*; *Benchenane et al., 2010*; *Fujisawa and Buzsáki, 2011*) and greater information transfer between mPFC and hippocampal neurons (*Ito et al., 2018*; *Padilla-Coreano et al., 2019*). Phase-locking to hippocampal theta is also prevalent among cells in many other regions, including the entorhinal cortex (EC), amygdala, parietal cortex, thalamic nucleus reuniens, and some subcortical and brainstem nuclei (*Kocsis and Vertes, 1992*; *Sirota et al., 2008*; *Fujisawa and Buzsáki, 2011*; *Bienvenu et al., 2012*; *Fernández-Ruiz et al., 2017*; *Ito et al., 2018*). Theta phase-synchronization could thus be a general mechanism for relaying information between the hippocampus and a broad network of memory-related regions.

In humans, macroelectrode LFP recordings in epilepsy patients have revealed sporadically occurring theta oscillations in the hippocampus and cortex during spatial navigation and episodic memory engagement (*Ekstrom et al., 2005*; *Watrous et al., 2011*; *Lega et al., 2012*; *Watrous et al., 2013a*; *Zhang and Jacobs, 2015*; *Vass et al., 2016*; *Aghajan et al., 2017*; *Stangl et al., 2021*). Macroscale theta phase-synchronization within the MTL and PFC has consistently correlated with better memory performance (*Babiloni et al., 2009*; *Watrous et al., 2013b*; *Solomon et al., 2017*; *Zheng et al., 2019*; *Kunz et al., 2019*). Considerably less is known about how oscillations relate to neuronal firing in humans than in rodents. An early study in epilepsy patients found that a large percentage of MTL and neocortical neurons phase-locked to locally recorded theta oscillations (among other frequencies) as subjects navigated through a virtual environment (*Jacobs et al., 2007*), and another study found that MTL neurons phase-locked more strongly to locally recorded theta oscillations while subjects viewed images that they later recognized than those that they forgot (*Rutishauser et al., 2010*). These findings indicate that neural activity within the human episodic memory system is organized in part by a theta phase code. However, few studies in humans have examined interregional relations between spiking and LFP oscillation phase. A recent study found that increased coupling between spikes and distal theta oscillations in the MTL during an associative image encoding task predicted better subsequent recognition (*Roux et al., 2022*). Yet it remains unclear if distal spike–LFP interactions are mediated

**Table 1.** Neurons by region.

Table shows how many subjects had at least one neuron in each brain region, how many neurons were recorded in each region, and the median, lower-, and upper-quartile firing rates for these neurons.

| Region | Subjects | Neurons | Firing rate (Hz) |
|---|---|---|---|
| Hippocampus | 27 | 391 | 1.6 (0.6, 4.7) |
| Entorhinal cortex | 19 | 341 | 2.3 (1.0, 5.5) |
| Amygdala | 23 | 439 | 1.5 (0.6, 3.7) |
| Parahippocampal gyrus | 15 | 217 | 2.2 (0.8, 4.5) |
| Superior temporal gyrus | 5 | 139 | 3.4 (1.4, 8.6) |
| Orbitofrontal cortex | 15 | 193 | 2.0 (0.9, 4.9) |
| Anterior cingulate cortex | 8 | 134 | 3.1 (1.4, 6.8) |
| *Total* | *28* | *1854* | *2.0 (0.8, 5.0)* |

Research article

by, or occur independently of, phase-locking to locally recorded oscillations. Regional differences in phase-locking prevalence to hippocampal oscillations are also underexplored. To address these questions, we leveraged the rare opportunity to record single- and multi-neuron activity simultaneously with LFP oscillations in multiple brain regions, including the hippocampus, in 28 neurosurgical patients implanted with intracranial electrodes.

## Results

Subjects were implanted with depth electrodes in the hippocampus, EC, amygdala, parahippocampal gyrus (PHG), superior temporal gyrus (STG), orbitofrontal cortex (OFC), and anterior cingulate cortex (ACC). From microwires that extended from the tips of these depth probes, we recorded extracellular spikes from 1,854 single- and multi-units (hereafter called 'neurons' *Table 1*) as subjects navigated through a virtual environment while completing one of several spatial memory tasks whose data we pooled for this analysis (see 'Materials and methods'). In total, we identified 10–71 (median = 30.0) neurons per session across 55 recording sessions, and the firing rates of these neurons were log-normally distributed (median = 2.0 Hz). In addition, every subject had at least one microwire bundle implanted in the hippocampus, permitting neuronal firing to be analyzed simultaneously with oscillatory activity in the hippocampal LFP.

### Identifying oscillations in hippocampal microwire LFPs

Earlier studies that have reported oscillatory properties of the human hippocampus during navigation have primarily utilized implanted macroelectrodes that integrate activity over hundreds of thousands of neurons (*Ekstrom et al., 2005*; *Watrous et al., 2011*; *Aghajan et al., 2017*; *Vass et al., 2016*). As the microwires used in the present study record at far smaller spatial scales, we first considered whether microwires exhibit oscillatory properties comparable to those observed in macro-electrode LFPs. We focused on 1–30 Hz signals for this analysis, avoiding higher frequencies at which spike-related artifacts can complicate LFP interpretation (*Manning et al., 2009*; *Buzsáki et al., 2012*; *Ray, 2015*). Many individual electrodes showed peaks in spectral power that rose above the background 1/f line in session-averaged LFP spectrograms (*Figure 1A*), indicating the potential presence of oscillatory activity (*Donoghue et al., 2020*). The frequency and magnitude of these spectral peaks varied considerably across subjects (compare *Figure 1A* subpanels) yet appeared nearly exclusively between 2–20 Hz.

To determine if spectral peaks were associated with sustained oscillations versus asynchronous, high-amplitude events, we used the BOSC (Better OSCillation) detection method to identify time-resolved oscillatory 'bouts' in each hippocampal microwire recording (*Whitten et al., 2011*). Briefly, BOSC (alternatively called '$P_{episode}$') defines an oscillatory bout according to two threshold criteria: Spectral power at a given frequency must exceed (1) a statistically defined amplitude above the 1/f spectrum, for (2) a minimum defined duration (we used 3 cycles; see 'Materials and methods' for more

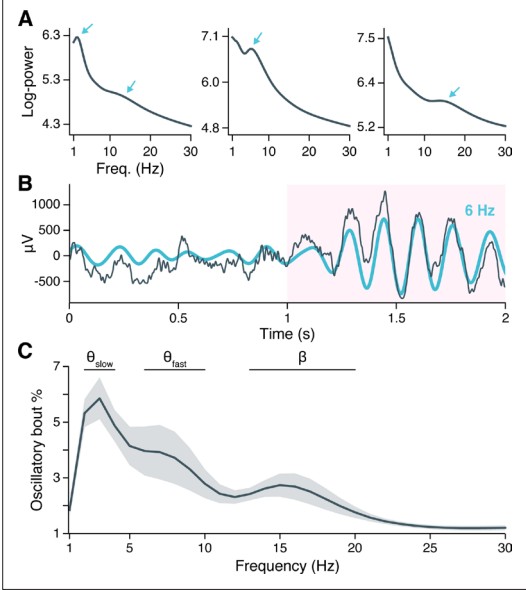

**Figure 1.** Neural oscillations in the hippocampus. (**A**) Spectral power across the recording session is shown for hippocampal local field potentials (LFPs) from three example subjects. Arrows indicate spectral peaks above the background 1/f spectrum. (**B**) A hippocampal LFP trace (gray line = raw LFP, cyan line = 6 Hz–filtered LFP) is shown immediately before and during a Better OSCillation (BOSC)–detected theta oscillation, highlighted in pink. (**C**) Mean ± SEM percent time, across 28 subjects, that BOSC-detected oscillations were present in hippocampal LFPs at each frequency from 1 to 30 Hz.

The online version of this article includes the following figure supplement(s) for figure 1:

**Figure supplement 1.** Neural oscillations outside the hippocampus.

**Figure supplement 2.** Oscillatory bout co-occurrence and waveform asymmetry at 3 Hz, 7 Hz, and 15 Hz.

details). ***Figure 1B*** shows an example hippocampal LFP in which an initially aperiodic, '1/f-like' signal transitioned into a strong, 6 Hz oscillation that persisted for 6 cycles, with the BOSC-defined oscillatory bout highlighted in pink.

Across subjects, hippocampal oscillatory bouts were present ~1–6% of the time at the examined frequencies (***Figure 1C***; ***Figure 1—figure supplement 2*** shows oscillatory prevalence in other regions for comparison). The prevalence of these oscillations was not uniform across frequencies, but instead clustered around three, well-separated bands with peaks at 3 Hz, 7 Hz, and 15 Hz. These frequencies are consistent with the hippocampal slow theta (alternatively 'delta' 2–4 Hz), fast theta (6–10 Hz), and beta band rhythms (13–20 Hz) previously described in macroelectrode recordings, and the prevalence of oscillatory bouts in our data was comparable to these earlier studies (***Ekstrom et al., 2005***; ***Lega et al., 2012***; ***Watrous et al., 2013a***; ***Goyal et al., 2020***).

As peaks at 3 Hz, 7 Hz, and 15 Hz could reflect harmonic resonance or waveform asymmetries of a single oscillation, we sought to verify whether oscillatory bouts at these frequencies occurred independently. Mean LFP waveforms during the first three cycles of each oscillatory bout showed symmetrical, sinusoidal shapes at each peak frequency, without any apparent harmonics (***Figure 1—figure supplement 1A***). We computed an asymmetry index for each waveform and confirmed that, on average across subjects, the 3 Hz and 7 Hz oscillations were nearly perfectly symmetrical, while the 15 Hz oscillation showed a very small asymmetry associated with a longer (by <1 ms) ascending than descending period (***Figure 1—figure supplement 1B***). Finally, we examined the extent to which oscillatory bouts at each peak frequency occurred at overlapping times, measuring the Dice similarity coefficient between oscillatory bouts at each peak frequency and all remaining frequencies. We found that the 3 Hz, 7 Hz, and 15 Hz oscillations occurred at largely separable times (***Figure 1—figure supplement 1C***). We concluded that hippocampal oscillatory bouts occur in three independent bands, centered at 3 Hz, 7 Hz, and 15 Hz.

Oscillatory prevalence varied between these frequency bands ($\chi^2(2) = 13.9$, $p < 0.0001$, likelihood ratio test between linear mixed-effects models testing frequency band as a fixed effect and holding subject as a random effect), such that slow theta was more prevalent than fast theta ($z = 2.4$, $p = 0.0336$, post-hoc pairwise $z$-tests, Bonferroni-Holm–corrected for multiple comparisons) or beta oscillations ($z = 3.9$, $p = 0.0002$), while fast theta and beta oscillations occurred at similar rates ($z = 1.5$, $p = 0.1218$). These findings indicate that the human hippocampus exhibits several distinct, low-frequency oscillations that are conserved across spatial scales spanning several orders of magnitude, from microwire to macroelectrode fields. Moreover, theta oscillations are the predominant oscillatory component of the hippocampal LFP during virtual navigation.

## Individual neuron phase-locking to hippocampal oscillations

Having confirmed the presence of hippocampal theta and beta oscillations, we next asked how these oscillations interacted with the timing of neuronal firing throughout recorded regions (***Table 1***). We quantified the phase-locking strength of individual neurons to ipsilateral hippocampal oscillations at a range of frequencies, 1–30 Hz. A neuron's phase-locking strength was defined as the mean resultant length (MRL) of hippocampal LFP phases across spike times at a given frequency, z-scored against a null distribution of MRLs obtained by circularly shifting the neuron's spike train 10,000 times at random (see 'Materials and methods'). To control for the possibility that some neurons might phase-lock to asynchronous events in the hippocampal LFP, such as sharp waves or interictal discharges (***Skelin et al., 2021***; ***Reed et al., 2020***), we restricted our analysis to spikes that coincided with BOSC-detected oscillatory bouts at each frequency, excluding 11% of neurons for which the number of included spikes did not suffice to accurately gauge phase-locking (see 'Materials and methods').

***Figure 2A*** illustrates the phase-locking of an EC neuron whose spikes appear in raster format above a simultaneously recorded, 3 s hippocampal LFP trace exhibiting slow theta rhythmicity. The neuron fired in bursts of 2–8 spikes on a majority of theta cycles, with each burst generally aligned with the theta cycle peak, while nearly no spikes occurred near the theta trough. Next, we examined the population phase-locking statistics for this neuron across the recording session (***Figure 2C***). Computing the mean hippocampal LFP trace surrounding each spike (the 'spike-triggered average LFP'), we confirmed that the neuron preferentially fired just after the theta peak, with synchronous theta oscillations extending more than a full cycle before and after spike onset (***Figure 2C***, left subpanel, blue line). As a control, we also examined a spike-triggered average LFP drawn at random from the null

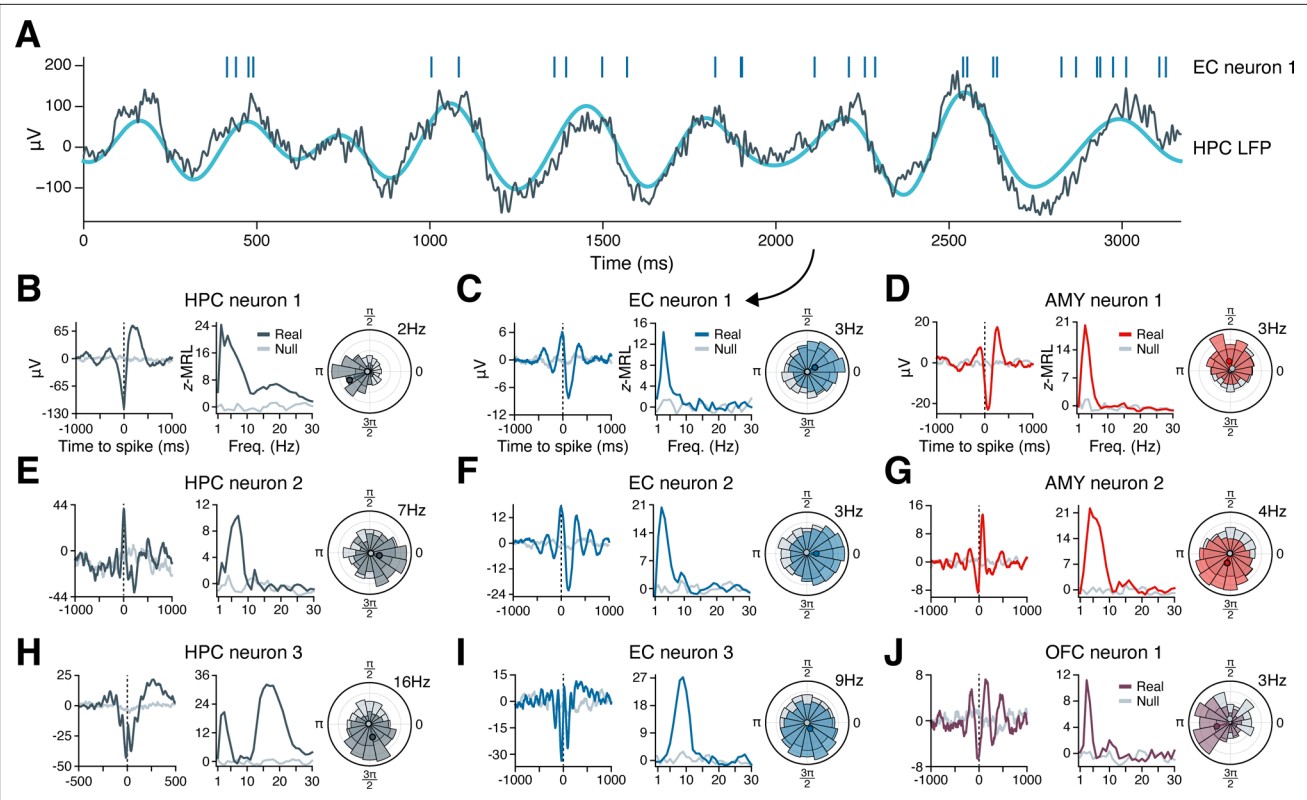

**Figure 2.** Example phase-locking to hippocampal oscillations. (**A**) Spikes from an EC neuron (top, vertical lines) are shown alongside local field potential (LFP) activity in the hippocampus during a slow theta oscillation (gray line = raw LFP, cyan line = 3 Hz–filtered LFP). Panel (**C**) shows phase-locking statistics for this neuron across the recording session. (**B–J**) Shown are nine neurons in the HPC (left column), EC (middle column), AMY (right column, top two rows), and OFC (right column, bottom row) that phase-locked to oscillatory signals in the hippocampus while subjects navigated through a virtual environment. The left subpanel for each neuron shows the mean hippocampal LFP centered on the time of each spike. The middle subpanel shows the phase-locking strength at each frequency relative to a null distribution of circularly shifted spikes. The right subpanel shows the spike–phase distribution at the maximum phase-locking frequency. Dark gray (HPC), blue (EC), red (AMY), and purple (OFC) lines correspond to true spike times, while light gray lines correspond to circularly shifted spike times from a single draw from the null distribution. HPC = hippocampus; EC = entorhinal cortex; AMY = amygdala; OFC = orbitofrontal cortex.

distribution, which showed a nearly flat line consistent with absent phase-locking (*Figure 2C*, left subpanel, gray line). Graphing this neuron's phase-locking strength at frequencies from 1 to 30 Hz revealed that phase-locking to hippocampal oscillations occurred only in the slow theta band, peaking at 3 Hz (*Figure 2C*, middle subpanel). Finally, the circular histogram of spike-coincident, 3 Hz hippocampal LFP phases showed that most spikes occurred within a quarter-cycle after the theta peak (*Figure 2C*, right subpanel). *Figure 2B and D–J* applies this analysis to representative neurons in the hippocampus, EC, amygdala, and OFC that phase-locked to LFP oscillations in the hippocampus. Most neurons exhibited unimodal peaks in phase-locking strength, most commonly in the theta range.

## Regional differences in hippocampal phase-locking

We next examined phase-locking at the population level, first considering the percentage of neurons in each region that significantly phase-locked to ipsilateral hippocampal LFP oscillations, irrespective of frequency. For each neuron, we derived an empirical phase-locking *p*-value by comparing the neuron's maximum phase-locking strength, across frequencies, to its null distribution of maximum phase-locking strengths (see 'Materials and methods'). We then applied false discovery rate (FDR) correction at $\alpha = 0.05$ to the distribution of *p*-values within each region. Finally, for each region outside the hippocampus, we performed the same analyses and statistical corrections with respect to LFP oscillations in each neuron's local region, proximal to the electrode from which a neuron was recorded. This last step allowed us to directly compare phase-locking rates to local versus remote hippocampal oscillations.

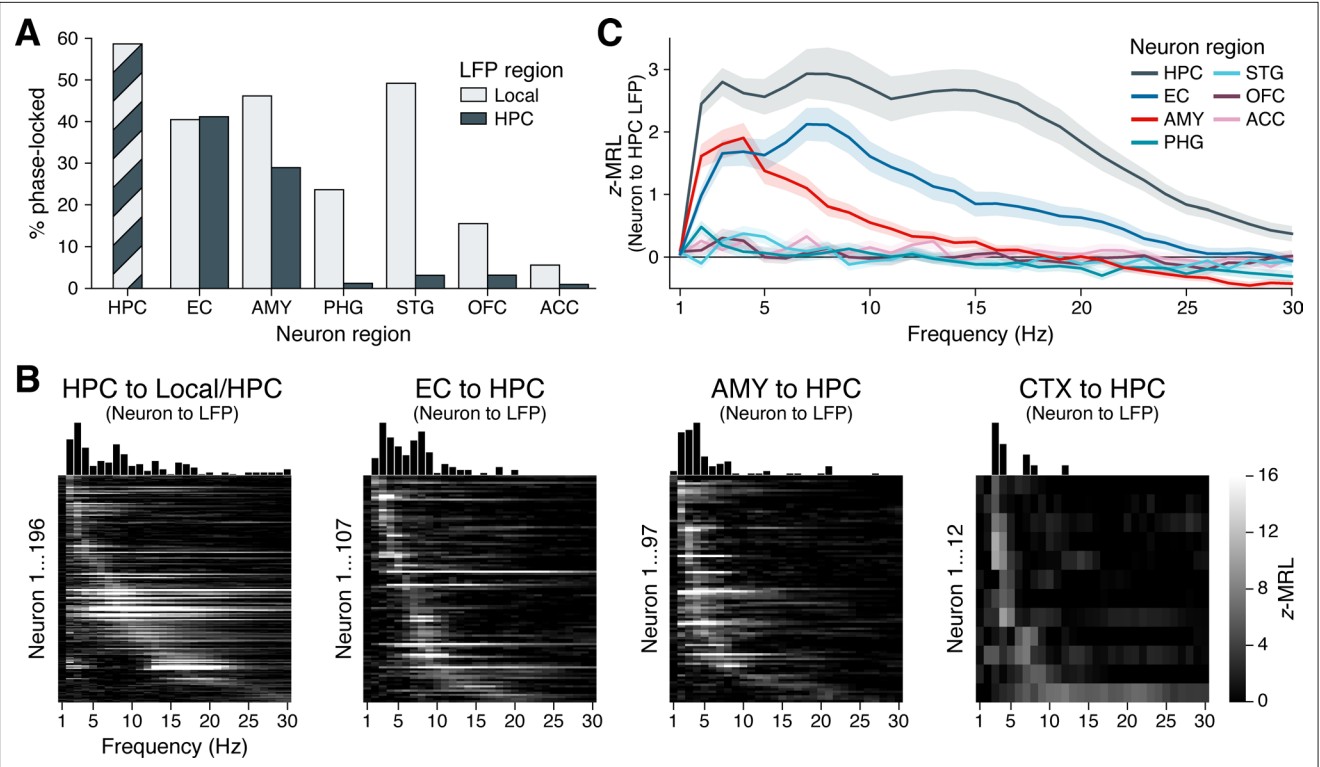

**Figure 3.** Phase-locking to hippocampal oscillations by region and frequency. (**A**) Bars show the percentage of neurons in each region that phase-locked to locally recorded local field potential (LFP) oscillations (light gray) and hippocampal LFP oscillations (dark gray). (Note that local and hippocampal LFP is identical for hippocampal neurons.) Phase-locking significance was set at false discovery rate (FDR)–corrected $p < 0.05$ within each bar group. (**B**) Heatmaps show the phase-locking strength (z-MRL; color scale intensity) by hippocampal LFP oscillation frequency (x-axis) for all significantly phase-locked neurons (y-axis; each row = one neuron) in the HPC, EC, AMY, and remaining regions (CTX), respectively. Neurons in each region are sorted from top to bottom by frequency of maximum phase-locking strength. Neurons depicted match the dark gray bars in (**A**). (**C**) Mean ± SEM phase-locking strength by hippocampal oscillation frequency is shown for all neurons in each region, regardless of their individual phase-locking significance as depicted in (**A**) and (**B**). HPC = hippocampus; EC = entorhinal cortex; AMY = amygdala; PHG = parahippocampal gyrus; STG = superior temporal gyrus; OFC = orbitofrontal cortex; ACC = anterior cingulate cortex.

The online version of this article includes the following figure supplement(s) for figure 3:

**Figure supplement 1.** Phase-locking to local oscillations.

*Figure 3A* illustrates these analyses. As expected, neurons within the hippocampus phase-locked to hippocampal oscillations at the highest rate among recorded regions, with 59% of hippocampal neurons significantly phase-locked after FDR correction. High phase-locking rates to the hippocampus were also found for neurons in the EC (41%) and amygdala (29%), with phase-locking rates in the EC significantly higher than those in the amygdala ($z = 3.6$, $p = 0.0004$, post-hoc pairwise $z$-test from a logistic mixed-effects model testing neuron region as a fixed effect and holding subject as a random effect). Whereas amygdala neurons phase-locked to local oscillations at significantly higher rates (46%) than to oscillations in the hippocampus ($\chi^2(1) = 32.6$, $p < 0.0001$), neurons in the EC phase-locked to local (40%) and hippocampal oscillations at indistinguishable rates ($\chi^2(1) = 0.2$, $p = 0.6672$, likelihood ratio tests between logistic mixed-effects models testing oscillation region as a fixed effect and holding subject as a random effect).

These results stood in stark contrast to all remaining regions, where phase-locking to the hippocampus occurred at rates below 5%. Phase-locking to local oscillations was nonetheless prevalent in the PHG (24%) and STG (49%), indicating that many of these neurons fired at specific phases of LFP oscillations — just not those recorded in the hippocampus. In two regions of the prefrontal cortex, local phase-locking rates were relatively low (16% of OFC neurons and 6% of ACC neurons) although still significantly higher than phase-locking rates to the hippocampus (OFC: $\chi^2(1) = 20.9$, $p < 0.0001$; ACC: $\chi^2(1) = 5.6$, $p = 0.0178$; likelihood ratio tests between logistic mixed-effects models, as above). Altogether, these results highlight a triad of regions — the hippocampus, EC, and amygdala — that

features strong spike-time synchronization to hippocampal oscillations, while neurons in more remote, cortical regions that are known to interact with hippocampus-dependent processes (*Eichenbaum, 2000*; *Squire, 2011*; *Ranganath and Ritchey, 2012*) phase-locked minimally to hippocampal rhythms.

## Frequencies of hippocampal phase-locking

Individual neuron examples suggested that phase-locking to the hippocampus occurred most commonly at theta frequencies (*Figure 2*), although our analysis of hippocampal LFPs revealed oscillations extending up to ~20 Hz (*Figure 1*). Does this observation of preferential theta phase-locking hold at the population level, and does the frequency of hippocampal phase-locking vary by a neuron's region of origin? To answer these questions, we generated heatmaps of phase-locking strength by frequency for all neurons that phase-locked significantly to hippocampal oscillations at *any* frequency, as defined in the previous section (*Figure 3B*; these neurons correspond to the dark gray bars in *Figure 3A*). We made separate heatmaps for neurons in the hippocampus, EC, amygdala, and remaining regions, sorting the neurons in each region by frequency of maximum phase-locking strength. *Figure 3—figure supplement 1* shows analogous heatmaps for neurons in each region with respect to local, rather than hippocampal, oscillations, matching the population of neurons represented by the light gray bars in *Figure 3A*.

In the hippocampus, neurons phase-locked to local oscillations predominantly between 2–20 Hz. Only a few neurons phase-locked weakly at higher frequencies, which may be largely attributable to false discoveries (*Figure 3B*, far-left subpanel). Within the 2–20 Hz range, phase-locking was not unimodal, but instead clustered around three distinct peaks in the slow theta, fast theta, and beta bands. Most hippocampal neurons phase-locked only to a single band, with the exception of neurons that phase-locked maximally to beta oscillations, which also showed a near-universal tendency to phase-lock strongly to slow theta (see for example *Figure 2H*). These neurons may best be classified as nested slow theta × beta phase-locking neurons, which to our knowledge have not previously been reported. In contrast, we did not observe nested phase-locking between fast theta and beta oscillations or between any other pair of frequency bands.

Among neurons outside the hippocampus, phase-locking to hippocampal oscillations occurred within a more constrained frequency range, between 2–10 Hz (*Figure 3B*, right three subpanels). In the EC, similar numbers of neurons showed preferential phase-locking to slow and fast hippocampal theta, respectively. In the amygdala and remaining cortical regions, this balance shifted: Only a few neurons phase-locked to fast hippocampal theta, while most neurons coupled exclusively to slow theta. Thus, while hippocampal neurons phase-locked to both theta and beta bands, for neurons outside the hippocampus, spike-time synchronization with hippocampal oscillations was restricted to theta frequencies.

We confirmed these conclusions in a secondary analysis that examined the mean phase-locking strength at each frequency across all neurons in each region, regardless of individual phase-locking significance (*Figure 3C*). This approach benefited from not requiring an explicit significance threshold to be defined. Instead, we assumed that if the neurons in a given region *did not* phase-lock measurably to the hippocampus, then the mean phase-locking strength across these neurons would approach zero with increasing sample size, since they would exhibit no difference against the null distribution. Indeed, population phase-locking strengths were close to zero across frequencies for neurons in the PHG, STG, OFC, and ACC, consistent with the relative absence of individually phase-locked neurons in these regions. In contrast, neurons in both the EC and the amygdala phase-locked strongly to slow hippocampal theta frequencies, while neurons in the EC, but not the amygdala, exhibited a secondary rise in phase-locking strength to fast hippocampal theta. Finally, neurons in the hippocampus showed stronger phase-locking to hippocampal oscillations at all frequencies than neurons in any other region, with peaks in phase-locking strength at all three oscillatory bands: slow theta, fast theta, and beta.

## Local oscillation effects on remote hippocampal phase-locking

Our data reveal that neurons not only within the hippocampus, but in remote regions — particularly the entorhinal cortex and amygdala — phase-lock to hippocampal theta oscillations. How do these remote spike–phase associations occur? One possibility, given the strength of phase-locking to local oscillations (*Figure 3—figure supplement 1*), is that phase-locking to the hippocampus is an indirect phenomenon, facilitated by transient phase coupling between oscillations in different

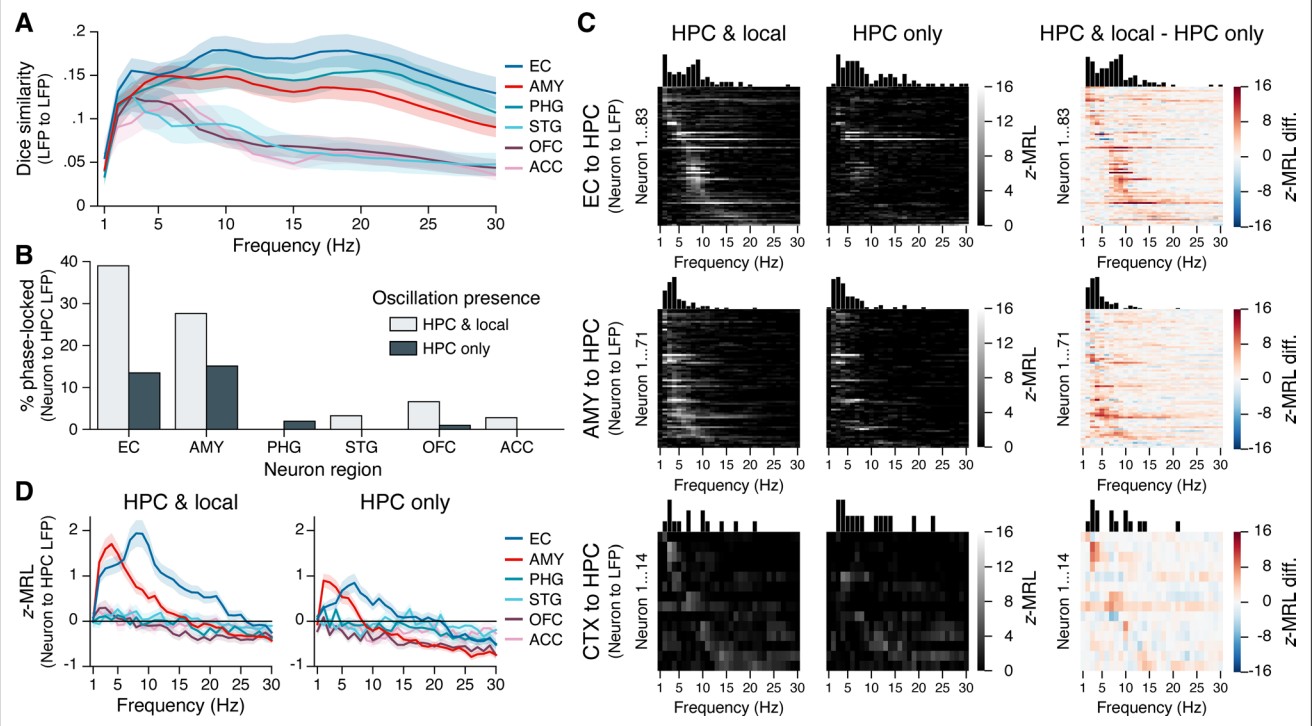

**Figure 4.** Phase-locking to hippocampal oscillations with and without co-occurring local oscillations. (**A**) Mean ± SEM (across 28 subjects) Dice coefficient across subjects shows the percent overlap between oscillatory bouts in the hippocampus and in each extrahippocampal region. (**B**) Bars show the percentage of neurons in each region that phase-locked to hippocampal oscillations when local oscillations were present (light gray) or absent (dark gray). Phase-locking significance was set at false discovery rate (FDR)-corrected $p < 0.05$ within each bar group. (**C**) Heatmaps show the phase-locking strength by hippocampal LFP oscillation frequency for all significantly phase-locked neurons in the EC (top row), AMY (middle row), and remaining regions (CTX; bottom row), when hippocampal and local oscillations co-occurred (left column) versus when only hippocampal oscillations occurred (middle column). The right column shows the left column minus middle column values. Neurons in each region are sorted from top to bottom by frequency with the maximum phase-locking strength, and the sorting order is constant across columns within each row. Neurons depicted match the union of light gray and dark gray bars in (**B**). (**D**) Phase-locking to the hippocampus is shown during co-occurring local and hippocampal oscillations (left) or only hippocampal oscillations (right). Each subpanel shows the mean ± SEM (across 28 subjects) phase-locking strength by hippocampal oscillation frequency for all neurons in each region, regardless of their individual phase-locking significance as depicted in (**B**) and (**C**). HPC = hippocampus; EC = entorhinal cortex; AMY = amygdala; PHG = parahippocampal gyrus; STG = superior temporal gyrus; OFC = orbitofrontal cortex; ACC = anterior cingulate cortex.

The online version of this article includes the following figure supplement(s) for figure 4:

**Figure supplement 1.** Two explanations for remote phase-locking to hippocampal theta.

**Figure supplement 2.** Phase-locking to local oscillations with and without co-occurring hippocampal oscillations.

regions (*Figure 4—figure supplement 1*, blue arrows). In rodents, however, neurons in some regions phase-lock to hippocampal theta even in the absence of a local theta rhythm (*Siapas et al., 2005*), suggesting that interregional oscillatory coupling is not a strict requirement for remote spike–phase associations (*Figure 4—figure supplement 1*, red arrow).

To examine how interregional oscillatory coupling contributed to remote spike–phase associations, we first considered the co-occurrence of oscillatory bouts in the hippocampus and in each extrahippocampal region. We reasoned that if remote spike–phase associations were mediated by oscillatory coupling, then regions where neurons phase-locked to the hippocampus at higher rates should also show higher levels of oscillatory co-occurrence. Consistent with this hypothesis, hippocampal oscillations overlapped more with oscillations in the EC and amygdala than with oscillations in the STG, OFC, and ACC at most frequencies (*Figure 4A*; overlap calculated using the Dice similarity coefficient). However, hippocampal and PHG oscillations also overlapped strongly despite the relative absence of PHG neuron phase-locking to the hippocampus (*Figure 3A*) and abundant PHG neuron phase-locking to local theta (*Figure 3—figure supplement 1*). Moreover, the overlap between local and hippocampal oscillations never exceeded 20% in any region at any frequency, indicating that

neurons could, in principle, phase-lock to hippocampal oscillations independent of local oscillations, and vice versa. Overall, these results provide a mixed view for the hypothesis that interregional oscillatory coupling and remote spike–phase associations are interchangeable.

Next, we directly compared how remote phase-locking to the hippocampus varied as a function of local phase-locking effects. For each extrahippocampal neuron, we divided spikes into two categories: (1) spikes that occurred when an oscillation was present in *both* the hippocampus and a neuron's local region, and (2) spikes that occurred when an oscillation was present in the hippocampus but *not* the neuron's local region. As chance-level phase-locking values depend on sample size, for each neuron we matched the number of spikes in each group, at each frequency, excluding neurons with insufficient sample size (<50 spikes at any frequency; see 'Materials and methods'). We then applied the same methods for determining phase-locking strength and significance as described in the previous section.

*Figure 4B* shows the results from these analyses. FDR-corrected phase-locking rates during co-occurring local and hippocampal oscillations were comparable to phase-locking rates when all spikes were included (see *Figure 3A*), with high phase-locking to hippocampal oscillations among neurons in the EC and amygdala and minimal phase-locking among neurons in other regions. In contrast, when hippocampal oscillations occurred without co-occurring local oscillations, phase-locking rates to the hippocampus declined by nearly two-thirds in the EC (from 39% to 14% of neurons) and by half in the amygdala (from 28% to 15%), while phase-locking to the hippocampus in other regions mostly vanished. Phase-locking strength to the hippocampus decreased specifically at theta frequencies, and even neurons that remained significantly phase-locked in the absence of local oscillations showed reduced phase-locking strength (*Figure 4C*). We also considered the converse question, asking whether phase-locking to local oscillations depended on the presence of co-occurring oscillations in the hippocampus. While hippocampal oscillation presence did not affect local phase-locking rates in the amygdala and neocortex, in the EC the percentage of locally phase-locked neurons was reduced by more than half when hippocampal oscillations were absent (*Figure 4—figure supplement 2*).

Finally, we confirmed these findings at the population level by computing the mean phase-locking strength across all neurons in each region, without regard to phase-locking significance, while still matching the number of spikes at each frequency between conditions in which local and hippocampal oscillations co-occurred, or in which only hippocampal oscillations occurred. As in *Figure 3C*, when local and hippocampal oscillations co-occurred, EC and amygdala neurons both phase-locked strongly to slow hippocampal theta, phase-locking to fast hippocampal theta was restricted to EC neurons, and other regions showed negligible phase-locking to hippocampal oscillations at any frequency (*Figure 4D*, left subpanel). When local theta was absent, the strength of EC and amygdala neuron phase-locking to hippocampal theta was reduced by half, while still remaining well above chance (*Figure 4D*, right subpanel). Collectively, these results provide direct evidence that interregional LFP–LFP theta coupling augments but is not strictly required for extrahippocampal neuron phase-locking to hippocampal theta.

## Discussion

By combining datasets of single- and multi-neuron recordings in human subjects, we provide an empirical test of the hypothesis that LFP oscillations in the hippocampus synchronize the timing of neuronal firing both within the hippocampus and in functionally associated regions. Consistent with prior studies, we identify sporadic, oscillatory bouts in hippocampal LFPs within slow theta (2–4 Hz), fast theta (6–10 Hz), and beta (13–20 Hz) bands while subjects engaged in virtual navigation. Individual hippocampal neurons phase-lock to oscillations in each of these bands, including a previously undiscovered group of neurons that phase-lock to nested slow theta and beta rhythms. Outside the hippocampus, phase-locking to hippocampal oscillations occurs in both a region-specific (primarily EC and amygdala neurons) and frequency-specific (theta-preferring) manner. We further show a dissociation between region and frequency in the selective phase-locking of EC neurons to fast hippocampal theta, whereas neurons in all regions outside the hippocampus show some level of phase-locking to slow hippocampal theta. Finally, we provide the first direct evidence in humans that LFP–LFP coupling enhances spike-time synchronization between regions, as extrahippocampal neurons phase-lock approximately twice as strongly to hippocampal theta when local theta oscillations co-occur, as when local theta is absent. Taken together, these findings reveal a fundamental relationship between MTL

neuron firing and hippocampal theta phase that underscores a hypothesized role for theta oscillations in routing the information contents of memory.

We note a particularly striking difference between phase-locking rates to hippocampal theta in the EC and amygdala (~30–40% of neurons) relative to all other recorded regions, which phase-lock minimally (<5%) despite their associations with hippocampus-dependent processes (*Eichenbaum, 2000*; *Squire, 2011*; *Ranganath and Ritchey, 2012*). This result is consistent with structural anatomy, as the hippocampus maintains strong, reciprocal connections with the EC and amygdala while connections to neocortex are sparser (*Amaral, 2011*). Still, given evidence in rodents that some mPFC neurons project directly to the hippocampus (*Rajasethupathy et al., 2015*), phase-lock to hippocampal theta (*Siapas et al., 2005*; *Hyman et al., 2005*; *Sirota et al., 2008*; *Ito et al., 2018*; *Padilla-Coreano et al., 2019*), and are critical for memory retrieval (*Rajasethupathy et al., 2015*; *Yadav et al., 2022*), we expected ACC and OFC neurons to show stronger associations with hippocampal theta than we observed. One possibility is that strong phase-locking to hippocampal theta occurs in the EC and amygdala at baseline, whereas phase-locking among neurons in the mPFC and other cortical areas is task-dependent. Consistently, a recent study in humans found that ACC and pre-supplementary motor area neurons phase-locked to hippocampal theta during a task-switching experiment in which subjects alternated between making recognition memory-based or categorization-based decisions (*Minxha et al., 2017*). It will be interesting for future work to consider how phase-locking rates vary by region under different task and stimulus conditions.

We find two differences in hippocampal phase-locking properties between the EC and amygdala. First, as in neocortical regions, amygdala neurons phase-lock at higher rates to local than to hippocampal oscillations, and local and hippocampal phase-locking occur at least somewhat independently. By contrast, EC neurons phase-lock to local and hippocampal theta oscillations at indistinguishable rates, and phase-locking is greatly disrupted when EC and hippocampal theta bouts are asynchronous. It is worth noting that in rodents, EC and hippocampal theta are phase-shifted but otherwise largely interchangeable, with EC inputs playing a major role in hippocampal theta generation (*Buzsáki, 2002*). Theta phase-synchronization between these regions is critical in explaining many circuit-level phenomena in rodents, including grid cell and place cell interactions, phase precession, and encoding/retrieval phase separation (*O'Keefe and Burgess, 2005*; *Hasselmo, 2005*; *Burgess et al., 2007*; *Bonnevie et al., 2013*; *Fernández-Ruiz et al., 2017*). Theta occurs more sporadically in humans and other primates than in rodents, and may differ between mammals in other ways not yet well understood (*Eliav et al., 2018*; *Trimper and Colgin, 2018*; *Bush and Burgess, 2019*). Still, our results indicate that as in rodents, EC and hippocampal neurons in humans retain a uniquely high degree of spike-time synchronization with an overlapping theta rhythm.

The second difference between EC and amygdala neurons concerns the frequency of hippocampal phase-locking, with neurons in both regions phase-locking to slow hippocampal theta but only EC neurons phase-locking measurably to fast theta. This result may be placed in context with recent observations that hippocampal theta frequency varies along the longitudinal axis of the hippocampus, with faster theta occurring more posteriorly (*Goyal et al., 2020*; *Penner et al., 2022*), where the density of EC relative to amygdalar afferents is greater (*Strange et al., 2014*). While most of our hippocampal electrodes were located anteriorly, precluding a direct analysis of EC and amygdala phase-locking by hippocampal electrode position, this hypothesis may be worth examining in a different dataset.

We note an important difference in our methodological approach compared to prior studies that examined spike–LFP phase relations in humans (*Jacobs et al., 2007*; *Rutishauser et al., 2010*; *Watrous et al., 2018*; *Kamiński et al., 2020*; *Minxha et al., 2020*; *Qasim et al., 2021*; *Roux et al., 2022*). These studies typically analyzed either all spikes or a large majority of spikes during time windows of interest, sometimes excluding spikes when spectral power fell below a predefined threshold — e.g., the bottom 25th percentile. Here, we wished to strictly test the hypothesis that neurons phase-lock to neural *oscillations* in the hippocampus, as defined by intervals when spectral power exceeds the 1/f spectrum by a significant amount for a sustained duration (*Whitten et al., 2011*; *Donoghue et al., 2022*). We considered this approach especially important given the sporadic nature of oscillatory bouts in human LFP recordings and the prevalence of asynchronous, high-power artifacts — interictal discharges (*Reed et al., 2020*), sharp-wave ripples (*Skelin et al., 2021*), duplicate spikes across channels (*Dehnen et al., 2021*), and movement or other non-neural artifacts that escape algorithmic detection. In our experience, phase-locking analyses that did not restrict spikes to verified oscillations

produced qualitatively similar group-level results as we report here, but included many individual cases of likely spurious phase-locking to non-oscillatory signals. This methodological difference might explain discrepancies between our results and earlier findings that hippocampal neurons phase-lock to local oscillations at a wider range of frequencies — e.g., 20–30 Hz — that we did not observe (*Jacobs et al., 2007*).

This study has several important limitations. First, all subjects had pharmacoresistant epilepsy, and we cannot rule out that some results might stem from pathological activity. However, we sought to attenuate this possibility by analyzing spikes only during oscillatory bouts, and we are encouraged by the general agreement between our results and those in rodents. A second limitation concerns the quality of unit isolation, as we recorded spikes from single microwires with limited ability to resolve spiking contributions from different neurons. Although some studies in humans have attempted to distinguish between single-units and multi-units and between excitatory and inhibitory neurons, unit quality metrics from microwires do not instill high confidence in the accuracy with which these distinctions can be made, while the potential for higher-quality unit recordings using tetrodes or Neuropixels may soon provide clarity with respect to differences specific to cell type (*Despouy et al., 2020*; *Chung et al., 2022*). In the meantime, we believe it is unlikely that this limitation would change any of our main conclusions, which do not depend on knowing if a unit is truly 'single' or a combination of several neighboring cells.

Still little is known about the relations between theta phase-locking and human cognition (*Herweg et al., 2020*). Prior studies have focused on the behavioral correlates of phase-locking to local theta rhythms within the MTL; according to one, for example, successful image encoding depended on theta phase-locking strength among hippocampal and amygdala neurons (*Rutishauser et al., 2010*), while another study found that MTL neurons can represent contextual information in their theta firing phase (*Watrous et al., 2018*). Here, we show that hippocampal theta oscillations also inform the timing of neuronal firing in regions beyond the hippocampus, positioning theta oscillations at the interplay between local circuit computations and interregional communication. In light of these results, brain and behavioral or physiological dissociations between local and interregional phase-locking, and between spike–LFP and LFP–LFP phase-synchronization, merit further investigation. Such analyses could unite findings from animal and human studies and advance a more mechanistic account of hippocampus-dependent processes across multiple scales, from single neurons to macroscopic fields.

# Materials and methods

## Participants

Subjects were 28patients with pharmacoresistant epilepsy who were implanted with depth electrodes to monitor seizure activity. Clinical teams determined the location and number of implanted electrodes in each patient. We conducted bedside cognitive testing on a laptop computer. Subjects completed one of two experiments (see 'Spatial navigation tasks'): Yellow Cab (18 subjects) or Goldmine (10 subjects). Demographic information was unavailable for Yellow Cab participants and is given below in aggregate for 11 Goldmine subjects, including the 10 analyzed subjects plus 1 pilot subject for whom technical problems prevented successful data collection.

| Sex | | Race | | Age | |
|---|---|---|---|---|---|
| Female | 7 | Asian | 1 | 20–25 | 5 |
| Male | 4 | Black | 1 | 25–31 | 1 |
| | | White | 6 | 32–37 | 2 |
| | | Unknown or Not Reported | 3 | 38–43 | 3 |

All testing was completed under informed consent. Institutional review boards at the University of California, Los Angeles, and the University of Pennsylvania approved all experiments. The number of the UCLA IRB protocol on which the Goldmine experiment was conducted is #10–000973.

## Spatial navigation tasks

We analyzed data from 55 recording sessions (1–4 sessions per subject, mean duration = 33.6 min). During each session, subjects played one of two first-person navigation games, Yellow Cab or

Goldmine, in which they freely explored a virtual environment and retrieved objects or navigated to specific locations. Previous studies have described the details of these experiments (*Ekstrom et al., 2003*; *Jacobs et al., 2010*; *Schonhaut et al., 2023*); for the present study, we pooled data across these studies to generate a large sample for conducting electrophysiological analyses. We analyzed intervals in which subjects could freely navigate through the virtual environment.

## Recording equipment

Each subject was implanted with 6–12 Behnke-Fried depth electrodes that feature macroelectrode contacts for clinical monitoring and 40 μm–diameter, platinum-iridium microwires for measuring microscale LFPs and extracellular action potentials (*Fried et al., 1999*). Electrode localizations were confirmed by the clinical team from post-operative structural MRIs or post-operative CT scans co-registered to pre-operative structural MRIs. Microwires were packaged in bundles of eight high-impedance recording wires and one low-impedance wire that served as the recording reference. Each microwire bundle was threaded through the center of a depth probe and extended 5 mm from the implanted end. As microwires splay out during implantation and cannot reliably be visualized on post-operative scans, electrode localizations are regarded with a ~5 mm radius of uncertainty that preclude analyses at the level of regional substructures or hippocampal layers or subfields. Microwire LFPs were amplified and sampled at 28–32 kHz on a Neuralynx Cheetah (Neuralynx, Tucson, AZ) or Blackrock NeuroPort (Blackrock Microsystems, Salt Lake City, UT) recording system.

## Spike sorting

We performed semi-automatic spike sorting and quality inspection on each microwire channel using the WaveClus software package in Matlab (*Quiroga et al., 2004*), as previously described (*Ekstrom et al., 2003*; *Schonhaut et al., 2023*). We isolated 0–8 units on each microwire channel, retaining both single-units and multi-units for subsequent analysis while removing units with low-amplitude waveforms relative to the noise floor, non-neuronal waveforms, inconsistent firing across the recording session, or other data quality issues. Spikes that clustered into separate clouds in reduced dimensional space were retained as separate units, while spikes that clustered into single clouds were merged. Repeated testing sessions occurred on different days, and we spike-sorted and analyzed these data separately.

## LFP preprocessing and spectral feature extraction

Microwire LFPs were downsampled to 1000 Hz, bandpass-filtered between 0.1–80 Hz using a zero-phase Hann window, and notch-filtered at 60 Hz to remove electrical line noise. Bandpass frequencies were selected to reduce signal drift at the low end and spike waveform artifacts (or other high-amplitude noise) at the high end, while maintaining sufficient distance from frequencies of interest for analysis. Lastly, we identified and removed a small number of dead or overly noisy channels, identified as those for which the mean, cross-frequency spectral power differed by >2 standard deviations from the mean spectral power across channels in each microwire bundle. The remaining LFP channels were manually inspected prior to further analysis as a secondary quality inspection step. Lastly, we extracted instantaneous spectral power and phase estimates for each preprocessed LFP channel by convolving the time domain signal with five-cycle complex wavelets at 30 frequencies, linearly spaced from 1 to 30 Hz.

## Oscillatory bout identification

For each LFP channel, we identified time-resolved oscillatory bouts at the 30 frequencies defined in the previous section using the BOSC (Better OSCillation) detection method, as described previously (*Whitten et al., 2011*). BOSC defines an oscillatory bout according to two threshold criteria: a power threshold, $P_T$, and a duration threshold, $D_T$. $P_T$ is set to the 95th percentile of the theoretical $\chi^2$ probability distribution of power values at each frequency, under the null hypothesis that powers can be modeled as a straight power law decaying function (the '1/f' spectrum). Defining $P_T$ for each frequency of interest requires first finding a best fit for 1/f. We obtained this fit by implementing the recently developed FOOOF (Fitting Oscillations & One-Over F) algorithm, which uses an iterative fitting procedure to decompose the power spectrogram into oscillatory components and a 1/f background fit (*Donoghue et al., 2020*). To avoid assuming that the 1/f spectrum was stationary across

the recording session, we divided the LFP into 30 s epochs and re-fit 1/f (and $P_T$, by extension) in each epoch. Finally, we set $D_T = 3/f$, consistent with the convention used in previous studies (*Ekstrom et al., 2005*; *Watrous et al., 2011*; *Aghajan et al., 2017*) that power at a given frequency $f$ must exceed $P_T$ for a minimum of three cycles for an oscillatory bout to be detected.

Oscillatory prevalence was calculated within three frequency bands of interest, defined as slow theta (2–4 Hz), fast theta (6–10 Hz), and beta (13–20 Hz). For each subject, we calculated the average oscillatory bout percentage across recording sessions, hippocampal microwire channels, and frequencies within each band. The resulting matrix provided a single measure of hippocampal LFP oscillation prevalence within each band, from each subject. Differences between bands were assessed using a linear mixed-effects model to account for repeated samples within subjects.

## Waveform asymmetry

Waveform asymmetry analyses were confined to oscillatory bouts as identified in the previous section. An inspection of the 3 Hz, 7 Hz, and 15 Hz oscillations averaged during the time windows corresponding to the first three cycles of each bout — 1000 ms, 428 ms, and 200 ms, respectively — qualitatively assessed asymmetries in these waveforms. Then, an asymmetry index was computed in keeping with previously established methods (*Roux et al., 2022*) for 3 Hz, 7 Hz, and 15 Hz waveforms. After initial preprocessing of the microwire LFPs (see 'LFP preprocessing and spectral feature extraction'), we applied a bandpass linear-phase Hamming-windowed FIR filter within a window of ±2 Hz centered at the frequency of interest, and identified local maxima and minima in windows equivalent to a half-cycle at this frequency. After aligning these extrema in the filtered LFP trace to the nearest peaks and troughs within a quarter-cycle in the raw, unfiltered LFP trace, we found the average difference between the time taken to ascend from a trough to the next peak and to descend from the peak to the subsequent trough. We normalized this average difference to the range $(-1, 1)$ by dividing by the cycle length $\frac{f_s}{f}$, where $f_s$ is the sampling frequency, and $f$ is the frequency of interest, giving the asymmetry index value. The asymmetry index values for each hippocampal recording were averaged first within subjects and then across subjects.

## Phase-locking strength and significance

We computed phase-locking strengths at 30 frequencies (1–30 Hz with 1 Hz spacing) between each neuron's spike times and oscillations in the hippocampus, as well as between each neuron's spike times and oscillations in the neuron's local region (other microwires in the same bundle, excluding the neuron's own recording wire due to spike contamination of the LFP). For both of these comparisons, we retained only spikes that coincided with BOSC-detected oscillatory bouts to avoid reporting spike–phase associations with non-oscillatory LFP phenomena. Phase-locking strength was then calculated as follows. First, at each frequency, we calculated the MRL of hippocampal LFP phases across spike times. The MRL is equal to the sum of phase angle unit vectors divided by the total number of samples, yielding a measure from 0 to 1 that indicates the extent to which the phase distribution is unimodal. This metric depends on sample size, with low $n$ yielding artificially high values due to chance clustering of phases. For this reason, we excluded neurons with <50 spikes at all frequencies of interest. Several other factors can artificially inflate the MRL, including nonuniform phase distributions in an underlying LFP signal, or autocorrelated spike times (*Siapas et al., 2005*). To control for these potential confounds, we used a permutation-based procedure in which we circularly shifted each neuron's spike train at random and then recalculated MRLs at each frequency, repeating this process 10,000 times per neuron to generate a null distribution. At each frequency, we then calculated phase-locking strength as the true MRL z-scored against null distribution MRLs at the same frequency.

To determine which neurons phase-locked significantly to local or hippocampal oscillations, we compared a neuron's maximum phase-locking strength across frequencies to a null distribution of maximum phase-locking strengths generated by taking the maximum of the null MRLs' z-scores across frequencies. We calculated an empirical p-value for each neuron with the formula $p = \frac{r + 1}{n + 1}$, where $r$ is the number of permuted values $\geq$ the true value for a given test statistic, and $n$ is the total number of permutations (*North et al., 2002*). Finally, we FDR–corrected p-values with the adaptive linear step-up procedure, which controls the expected proportion of true null hypotheses among rejected nulls for both independent and positively dependent test statistics, and has greater statistical power

than the commonly used Benjamini-Hochberg procedure (*Benjamini et al., 2006*). FDR correction was applied separately to *p*-values from each neuron region × LFP region (local or hippocampal) pair to control the expected proportion of false positives within each of these groups. Neurons with FDR-corrected $p < 0.05$ were deemed significantly phase-locked.

## Interregional oscillatory co-occurrence

Co-occurrence rates were determined between hippocampal and extrahippocampal oscillatory bouts by quantifying the Dice coefficient between each hippocampal electrode and each ipsilateral, extrahippocampal electrode. The Dice coefficient measures the similarity from 0 to 1 between two sets $A$ and $B$, with 0 indicating that the sets do not overlap and 1 indicating that $A$ and $B$ are equal: $Dice = \frac{2|A \cap B|}{|A| + |B|}$, where $|A|$ and $|B|$ correspond to the number of elements in each set and $|A \cap B|$ is the number of elements common to both sets. We calculated these values using binarized oscillation detection vectors (oscillation present or absent) as defined in 'Oscillatory bout identification,' separately at each 1–30 Hz frequency.

## Phase-locking to hippocampal oscillations during co-occurring or absent local oscillations

We divided spikes from each extrahippocampal neuron into two groups according to the following criteria: (1) BOSC-detected oscillations were present in both the hippocampus and a neuron's local region, or (2) BOSC-detected oscillations were present in the hippocampus but not the neuron's local region (*Figure 4*). These spike subsets were determined separately for each 1–30 Hz frequency. Phase-locking strengths were then calculated separately within each spike group, at each frequency, and significance determined relative to null distributions as described in 'Phase-locking strength and significance.' As chance-level phase-locking values depend on sample size, for each neuron we matched the number of spikes in each group, at each frequency, excluding neurons with insufficient sample size (<50 spikes at any frequency). For example, for neuron $i$ at frequency $j$, if 200 spikes occurred when local and hippocampal oscillations were both present and 150 spikes occurred when only hippocampal oscillations were present, we selected 150 spikes from the first group at random and proceeded to calculate phase-locking strength in each group. The same analytical approach was applied to a supplemental analysis (*Figure 4—figure supplement 2*) in which extrahippocampal spikes were subdivided as: (1) local and hippocampal oscillations were both present, or (2) local oscillations were present but hippocampal oscillations were absent.

## Statistics

Linear and logistic mixed-effects models with fixed slopes and random intercepts were performed using the lme4 package in R (*Baayen et al., 2008*). All models included a single random effect of subject and a single fixed effect of interest, as specified in each result. *p*-values were obtained from likelihood ratio tests between nested models (with versus without inclusion of the fixed effect). We adopted this approach to control for inter-subject differences in our data that conventional methods such as linear regression would overlook, as they assume independence between neurons. This approach was particularly important for comparing effects between regions, as each subject had electrodes in only a subset of the regions that we analyzed. For models in which the independent variable was a categorical measure with three or more levels, if the likelihood ratio test revealed a significant effect ($p < 0.05$), we performed post-hoc, pairwise *z*-tests on the fitted model terms with Bonferroni-Holm correction for multiple comparisons were noted in the Results.

## Software

Mixed-effects models were fit using the lme4 package in R (*Baayen et al., 2008*). Spike sorting was performed using the Wave_clus software package in Matlab (*Quiroga et al., 2004*). All additional analyses were performed, and plots were generated, using code that was developed in-house in Python 3, utilizing standard libraries and the following publicly available packages: astropy (*The Astropy Collaboration et al., 2022*), fooof (*Donoghue et al., 2020*), matplotlib (*Hunter, 2007*), mne (*Gramfort et al., 2013*), numpy (*Harris et al., 2020*), pandas (*McKinney, 2010*), seaborn (*Waskom, 2021*), scipy (*Virtanen et al., 2020*), statsmodels (*Seabold and Perktold, 2010*), and xarray (*Hoyer and Hamman, 2017*).

## Acknowledgements
We are grateful to the patients for their participation and thank the hospital staff and researchers who were involved in data acquisition. This work was supported by the National Science Foundation GRFP grant (DRS), NIH U01 (NS113198 to MJK) and NINDS (R01-NS033221 and R01-NS084017 to IF), and Deutsche Forschungsgemeinschaft (DFG) Grant HE 8302/1–1 (NAH).

## Additional information

### Funding

| Funder | Grant reference number | Author |
| --- | --- | --- |
| National Science Foundation Graduate Research Fellowship Program | | Daniel R Schonhaut |
| National Institutes of Health | 1U01NS113198-01 | Michael J Kahana |
| National Institute of Neurological Disorders and Stroke | R01-NS033221 | Itzhak Fried |
| National Institute of Neurological Disorders and Stroke | R01-NS084017 | Itzhak Fried |
| Deutsche Forschungsgemeinschaft | HE 8302/1-1 | Nora A Herweg |

The funders had no role in study design, data collection and interpretation, or the decision to submit the work for publication.

### Author contributions
Daniel R Schonhaut, Conceptualization, Data curation, Software, Formal analysis, Funding acquisition, Investigation, Visualization, Methodology, Writing - original draft, Writing – review and editing; Aditya M Rao, Formal analysis, Visualization, Methodology, Writing – review and editing, Aditya Rao was added as an author during the revision process in view of his contributions to the second round of revisions, for which he completed additional data analyses. The remaining authors are in agreement with inclusion and position in the author list; Ashwin G Ramayya, Conceptualization, Data curation, Methodology, Writing – review and editing; Ethan A Solomon, Nora A Herweg, Conceptualization, Methodology, Writing – review and editing; Itzhak Fried, Resources, Data curation, Funding acquisition, Writing – review and editing; Michael J Kahana, Conceptualization, Resources, Supervision, Funding acquisition, Writing – review and editing

### Author ORCIDs
Daniel R Schonhaut ⓘ http://orcid.org/0000-0001-8667-031X
Aditya M Rao ⓘ http://orcid.org/0000-0002-5854-5594
Ashwin G Ramayya ⓘ http://orcid.org/0000-0002-4444-0433
Nora A Herweg ⓘ http://orcid.org/0000-0002-4647-7408
Itzhak Fried ⓘ http://orcid.org/0000-0002-5962-2678
Michael J Kahana ⓘ http://orcid.org/0000-0001-8122-9525

### Ethics
Human subjects: All testing was completed under informed consent. Institutional review boards at the University of California, Los Angeles and the University of Pennsylvania approved all experiments. The number of the UCLA IRB protocol on which the Goldmine experiment was conducted is #10-000973.

### Decision letter and Author response
Decision letter https://doi.org/10.7554/eLife.85753.sa1
Author response https://doi.org/10.7554/eLife.85753.sa2

# Additional files

## Supplementary files
• MDAR checklist

## Data availability
The data used in this study is publicly available from the Cognitive Electrophysiology Data Portal. This dataset includes de-identified, raw EEG data, spike-sorted unit data, and preprocessed phase-locking data. Due to size constraints, the data can be accessed via a request form — requests will be evaluated to ensure the correct datasets are made accessible to those who request them. All data analysis code and JupyterLab notebooks can be freely downloaded from Zenodo.

The following dataset was generated:

| Author(s) | Year | Dataset title | Dataset URL | Database and Identifier |
| --- | --- | --- | --- | --- |
| Schonhaut DR, Rao AM, Ramayya AG, Solomon EA, Herweg NA, Fried I, Kahana MJ | 2024 | MTL neurons phase-lock to human hippocampal theta (code and data) | https://memory.psych.upenn.edu/Data | Cognitive Electrophysiology Data Portal, SchoEtal24 |

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
