## [Editor Report]

Large sample size electrophysiology in human brains is rare, and thus this work is an important contribution to the field. The mesoscopic descriptive analyses provide a convincing bridge to work done in other species and will likely further contribute to its long term value to the field.

---

## [Decision Letter]

**Decision letter after peer review:**

[Editors’ note: the authors submitted for reconsideration following the decision after peer review. What follows is the decision letter after the first round of review.]

Thank you for submitting your article "Single neurons throughout human memory regions phase-lock to hippocampal theta" for consideration by *eLife*. Your article has been reviewed by 3 peer reviewers, and the evaluation has been overseen by a Reviewing Editor and Laura Colgin as the Senior Editor. The following individual involved in review of your submission has agreed to reveal their identity: Antonio Fernandez-Ruiz (Reviewer #3).

The reviewers have discussed the reviews with one another and the Reviewing Editor has drafted this decision to help you prepare a revised submission.

Summary:

This is a very intriguing paper showing how hippocampal local field potentials couple with the activity of other cortical regions. This mechanism has been and continues to be extensively studied in other mammals, and thus its existence and relevance in humans is exciting. The reviewers were unanimous in their opinion that this work is worthy of publication in *eLife*, but pointed to key aspects of the paper that they felt were insufficient.

Essential revisions:

1. Hippocampal Data:

A) Critically, the authors should include similar analyses of hippocampal neurons (if they exist) and explain why they don't have them otherwise.

B) The nature of the theta oscillations needs to be explicated – are they short bursts accompanying eye movements or more sustained periods? – and example spectrograms and traces shown.

C) The authors should describe the anatomical locations of the hippocampal LFP electrodes, explain how they were chosen, and whether they varied from subject to subject.

2. New Analyses: The authors focus on linking hippocampal LFP and extra-hippocampal spikes. In addition to hippocampal LFP -> hippocampal spikes (point 1A), it would be valuable to assess hippocampal LFP -> extra-hippocampal LFP. This might include looking at dynamics of theta-band coherence and/or theta-modulation of high γ.

3. Data Analysis Issues:

A) While they explain that many of their spikes are coming from multi-units, they should be more upfront about this, and when possible, try to classify neurons as pyramidal or interneurons, and show results by neuron type / base firing rate.

B) Critically, there is concern that some of the analyses use the same set of data to e.g., pick frequency bands and analyze coupling, rather than using some form of cross-validation.

*Reviewer #1:*

Hippocampal theta oscillations are among the most prominent rhythms in the mammalian brain. Extensive research in rodents has shown that neurons not only within the hippocampus but in widespread cortical areas can be phase-locked to hippocampal theta. Such cross-regional communication within theta frames has been postulated to be the foundation of many hippocampal operations. While previous studies in humans have documented the relationship between LFP theta and spiking in the hippocampus, coupling between hippocampal LFP and more remote cortical areas have not been demonstrated in human subjects. This is the topic of the present work. The authors show that spikes of single (and mostly multiunit) neurons in multiple cortical regions both in the same and opposite hemipheres are phase locked to transient occurrence of hippocampal theta LFP in the 2-6 Hz range. However, phase-locking is stronger in structure know to be part of the 'limbic system', such as the amygdala and entorhinal cortex. Theta phase locking was stronger to hippocampal than to local LFP and the magnitude of spike phase locking increased when the power of theta increased, associated with increased high frequency power. The results are straightforward and the analysis methods are reliable. The novel information is limited but informative and documents a missing aspect of theta communication in the human brain.

Comments:

1. Given the simple message, the text is a bit long with many repetitions and loose ends. This applies to both Introduction and Discussion. Potential implications to learning, etc are interesting but the findings do not provide additional clues, thus those aspects of the discussion are mainly distractions. Instead, perhaps the authors would like to discuss potential mechanisms of remote unit entrainment. They are talking about multi-synaptic pathways but these are unlikely to be a valid conduit. Instead, the septum, entorhinal cortex or retrosplenial cortex, with their widespread projections, may be responsible for coordinating both hippocampal and neocortical areas.

2. Arguably, the weakest part of the manuscript is the lack of hippocampal neurons. The authors refer to their own previous papers, but in a story which compares hippocampal theta oscillations with remote unit activity, it is strange that the magnitude of theta phase-locking to local hippocampal neurons is not available for comparison.

3. How was the hippocampal LFP reference site chosen and did it vary substantially from subject to subject? Anterior or posterior locations?

4. The authors list 1233 single neurons but in the discussion they make it clear that most of them were multiple neurons. This should be emphasized up front and may be used as an excuse why the authors did not attempt to separate pyramidal cells from interneurons (interneurons have a much higher propensity to be entrained by projected rhythms).

5. Given that units were mixed, a logical extension would be to examine how hippocampal theta phase modulates high γ in neocortical areas. This could potentially yield a much larger data base, targeting the same question.

6. In the Discussion, the authors suggest that cross-regional theta phase coupling could be related to learning and other cognitive performance. However, spike-LFP coupling and coherence is confounded by LFP power increase and the authors cite Herweg et al., 2020 which did not find a relationship between theta power and memory performance. Is it then not logical to assume that cross-regional coupling may also not be related to memory?

7. Line 36. "Long-term potentiation and long-term depression in the rodent hippocampus are also theta phase-dependent (Hyman et al., 2003)." Pavlides et al. (Brain Res 1988) or Huerta and Lisman (Neuron 1995) are perhaps more relevant references here.

8. Line 82: "significant neocortical and contralateral phase-locking suggests". This is a strange phase. Perhaps significant phase locking of neurons in the neocortex in both hemispheres or similar would be a better formulation.

*Reviewer #2:*

In this study, Schonhaut et al., describe the phase locking statistics of cortical and subcortical neurons with respect to hippocampal local field potential (LFP) recorded in 18 epilepsy patients undergoing seizure monitoring. Nearly 30% of extrahippocampal neurons showed phase locking to some bandpassed hippocampal signal. Amygdalar and entorhinal neurons were more likely to be phase locked, as compared to neurons recorded in other neocortical sites. Most neurons showed the strongest phase locking to hippocampal theta (2-8 Hz), though neocortical and amygdalar neurons tended to phase lock to lower theta bands. Spikes that were phase locked to hippocampal rhythms occurred during local LFP-states that showed moderate correlations with the spectral patterns observed in the hippocampus. These data are interpreted within the broader "communication through coherence" hypothesis.

Large N, multi-region, single unit studies from humans are rare and the kind of mesoscopic descriptive analyses provided here serve as an important bridge between the large rodent literature on hippocampal physiology and human physiology and cognition. That said, there are some weaknesses in the analyses that could be addressed in a revised report. Also, a deeper discussion of the biological origin of human theta is merited in the discussion to address alternate explanations – beyond communication through coherence – of the data.

A similar statistical mistake was made several times. The author's logic goes like this: find the argmax in one sample, take the argument that generated that max, and use that to sample in another condition, and report that the max is higher in the first condition than the second. For example, on pg. 6 "This is difficult to reconcile with our results, in which 248/362 neurons (68.5%) phase-locked more strongly to hippocampal LFPs than to locally-recorded LFPs at their preferred hippocampal phase-locking frequency." The same flaw can be seen in Figure 5, where the spikes are sub-sampled to occur during strong phase locking in one condition, thus almost guaranteeing high power in the frequency bands that generated that strong phase locking (which was observed). This is a case in which cross-validating the data may be useful. The authors could take a subset of the hippocampal data to define the preferred frequency, and then test phase locking on the held out data from the hippocampus and cortex.

The relationship between power and phase locking is not fully controlled in this paper. The phase seems to be calculated irrespective of whether there is any instantaneous power at that frequency band, introducing noise. This will bias away from finding significant phase locking to frequency bands that occur transiently. Therefore, I recommend defining some threshold of the existence of the spectral signal prior to using that signal to calculate phase.

A related point has to do with the nature of the theta rhythm in the human. There has been considerable controversy over the years as to whether this is a comparable signal to that studied in the rodent. Based off the citations in this manuscript, and the nomenclature of the spectral band, the authors seek to make explicit the commonality of the underlying physiology, or function. Rodent theta is a sustained rhythm, while primate theta seems to come in bouts, perhaps even related to sampling statistics, such as saccades, leading to the suggestion that the apparent theta may be better thought of as semi-rhythmic evoked responses. How long were the bouts of high theta power? Was eye movement tracked? If so it would be important to relate the signal to eye movements. If the low frequency signal is phase locked to eye movement and potentially reflects semi-rhythmic information arriving to (from?) the hippocampus, then a stronger case could be made that hidden "third parties" synchronize the apparent communication through coherence observed here, and in fact there may be no communication at all.

The authors dedicate much of their discussion to relating the current result to the communication through coherence analyses. Oddly, LFP coherence was never addressed. A strong prediction of the current framing would be that: when coherence is high, phase locking should by high, and higher than other moments when power in either region is high but coherence is not observed. The authors should directly measure how phase locking is modulated by coherence.

The authors also lump together biological entities that should have difference phase locking behaviors. The amygdala is not a monolithic region, does phase locking differ by nucleus? Also, do fast spiking inhibitory cells differ from excitatory cells? The authors should relate their phase locking measure to mean firing rate to show that it is insensitive to lower level cell statistics. This is important since the conclusions of the study would be quite different if neurons in the entorhinal cortex had high rates which artifactually drove up phase locking values.

*Reviewer #3:*

The manuscript by Schonhaut et al. presents novel analysis on an impressive dataset of more than 1200 neurons across diverse brain areas in the human brain to investigate their modulation by hippocampal theta oscillations. They found a substantive proportion of cells phase-locked to hippocampal activity, mainly in the theta frequency band, in several areas know to be functionally related to the hippocampus, some of them receiving monosynaptic hippocampal inputs but other only indirect ones. These results extend previous reports in humans showing hippocampal interactions with these structures but at the level of mesoscopic activity and highlight the ubiquity of spike-theta timing and the importance of single-unit studies in humans. Additional analysis, detailed below, will contribute to give a better description of the data, provide stronger support for some of the authors' claims and clarify some issues.

1. I assume that the dataset also includes hippocampal units, why then excluding them from the analysis? Although the main novelty is in the coupling of cells in other structures with hippocampal LFPs, it would be useful to also compare it with the coupling of local hippocampal cells.

2. Include average power spectrum of hippocampal LFPs. Additional examples of raw LFP traces overlaid to spectrograms (perhaps in Supplementary) will help to illustrate the nature of hippocampal oscillations.

3. The authors compared fractions of significantly modulated units and their preferred frequencies across regions. While very informative, these analyses are not sufficient to capture the richness of spike-LFP interactions likely existing in the dataset. Were there differences in the strength of phase-locking across regions? (this analysis could be added to Figure 2). Studies in rodents have shown that theta phase-locked units in different structures have characteristics preferred firing phases (when hippocampal LFP is used as a reference). Authors can easily look if this is also the case in their data. They should include both pooled data statistics of mean phases across regions and single neuron examples of firing probability by LFP phase (such examples could be added to the single unit plots in Figure 1).

4. Did phase-locked and non phase-locked units have different properties? The authors can compare if they differ in basic properties such us mean firing rate, waveform width, inter-spike intervals, burstiness, etc., as it has been reported in other studies in non-human primates and rodents. These analyses could be extended to show if units with different properties also differ in their preferred phase-locking frequency, or phase. It would be very interesting if these analyses reveal the existence of heterogeneous cellular populations with different relation to hippocampal theta, even if the single-unit isolation quality is limited due to the low density recordings. In relation to this, authors should also plot unit auto-correlograms. ACGs can be computed for all the spikes, but also only for the strongly phase-locked spikes, to show if, at least during periods of strong oscillatory activity, some units show rhythmicity.

5. To better interpret the results in Figure 4, it would be important to know if the recording sites in both hippocampi were from the same sub-region and similar location along the longitudinal hippocampal axis in each subject and if the degree of synchrony between the LFP in both hemispheres. Coherence or phase-locking between LFPs across hemispheres should be computed and also power spectrum for both of them shown.

6. In Figure 4C-D it seems that phase-locking strength across hemispheres was not correlated but preferred frequency was. This should be quantified and mentioned in text before moving to the correlation in Figure 4E.

7. The analysis in Figure 5D should be complemented by also checking the LFP-LFP phase-locking between the local region and the hippocampus. Were periods of high LFP power correlation also reflect enhanced phase-phase coupling? Were the structures also more phase-synchronous during periods of stronger spike-LFP coupling? These analyses could provide a more direct support for the interpretation of the authors in line with the CTC hypothesis.

8. Was there any relation of the "strongly phase-locked" periods with global variables reflecting brain state (e.g. drowsiness versus attention to the task, etc.) or with the firing dynamics of the units (instantaneous firing rate or inter-spike intervals)?

[Editors’ note: further revisions were suggested prior to acceptance, as described below.]

Thank you for resubmitting your work entitled "MTL neurons phase-lock to human hippocampal theta" for further consideration by *eLife*. Your revised article has been evaluated by Laura Colgin (Senior Editor) and a Reviewing Editor.

The manuscript has been improved but there are some remaining issues that need to be addressed, as outlined below:

Essential revisions:

Please address the third reviewer's comments regarding (1) the directionality of spike/LFP coupling, (2) assessing whether different frequency bands can be explained as higher-order harmonics, and (3) adding a citation and potentially some discussion of the Roux 2022 paper.

*Reviewer #3 (Recommendations for the authors):*

1. At several parts in the manuscript the authors suggest that the coupling between extrahippocampal units to hippocampal oscillations is in the direction of LFP -> Spike (lines 168-169; 182-184; 238-239). I was surprised by this, because classically distal spike-LFP coupling is usually interpreted as directional coupling in the opposite direction (Spike -> LFP). This is because the common assumption would be that a if a spike in region A is coupled to the LFP in region B, then this is because the neuron in region A elicits post-synaptic currents in region B, hence spike -> LFP (see Buzsáki and Schomburg, 2015; Liebe et al., 2012; Jacob et al., 2018; see also Roux et al., 2022 for an example in the human MTL). Now, this doesn't mean that coupling in the other direction LFP ◊ Spike is not also possible, but it would require additional steps, whereby the LFP in region A entrains local neurons which then project to region B, where they elicit an LFP in region B which then induces firing of local neurons at specific phases. However, such an explanation is less parsimonious and thus requires additional evidence. Specifically, two additional analyses would be useful in this regard. First, the LFPs in region A and region B would have to be phase coupled. The authors demonstrate a coupling in terms of power, by measuring co-occurrence of 'bouts', but they do not demonstrate whether there is a consistent phase-relationship between the two regions where 'bouts' co-occur. Second, coupling measures which allow inferences about directionality should be applied to the spike and LFP time series. This can be done, for instance, by convolving the spike time series with a gaussian envelope and then applying the phase slope index to both time series (as done in Roux et al., 2022).

2. Oscillatory bouts appear in three frequencies, ~3 Hz, ~7 Hz, ~ 15 Hz, with decreasing density of occurrence (i.e. 3 > 7 > 15 Hz). This could be indicative of an asymmetric 3 Hz oscillation which induces spurious signals at the first and second harmonic. Indeed, the 15 Hz oscillation appears to be coupled to the 3 Hz oscillation which would be consistent with this assumption (although the fact that 7 Hz is not, would not support this argument, but still). To counter this the authors should show how often bouts in the three frequency windows co-occur. If the higher frequencies are a reflection of asymmetric wave shapes then there should be a tight correlation with respect to when the bouts occur. Furthermore, the authors could investigate the waveshape of the 3 Hz oscillation by calculating an asymmetry index (as done in Roux et al., 2022; see Figure 5, figure supplement 1).

3. A recent paper that seems highly relevant to the current one is not mentioned. I am referring to the paper by Roux et al. 2022 who also investigated local and distal (cross-regional) spike-LFP coupling in the human MTL during a memory task. In line with the current study, Roux et al. also show distal coupling of MTL neurons in theta. Furthermore, they demonstrate that this coupling is related to memory whereby coupling at faster frequencies predicts successful formation of associations, whereas coupling to slow frequencies predicts unsuccessful formation of associations. Crucially, and central to the current study, Roux et al. 2022 demonstrate that coupling at theta frequencies was correlated with the latency of co-firing of pairs of distally coupled neurons. Therefore, several statements in the paper should to be revisited in light of that previous study (i.e. lines 398-400; 311-316; 66-75). I assume the reason for why the authors did not include that study is that it appeared late in 2022, likely when this paper was already submitted or close to submission. I also think that the Roux et al. paper does not take away anything in terms of novelty from this paper, as Roux et al. were not able to split up the distal coupling into the different MTL subregions due to a lower yield in neurons. If anything, the two papers nicely complement each other and both support a central role of MTL theta oscillations in routing information in the human brain in the service of memory and navigation.

References:

Buzsáki G, Schomburg EW. 2015. What does γ coherence tell us about inter-regional neural communication? Nature Neuroscience 18:484-489. DOI: https://doi.org/10.1038/nn.3952, PMID: 25706474

Liebe S, Hoerzer GM, Logothetis NK, Rainer G. 2012. Theta coupling between V4 and prefrontal cortex predicts visual short-term memory performance. Nature Neuroscience 15:456-462,. DOI: https://doi.org/10.1038/nn. 3038, PMID: 22286175

Jacob SN, Hähnke D, Nieder A. 2018. Structuring of Abstract working memory content by fronto-parietal synchrony in primate cortex. Neuron 99:588-597.. DOI: https://doi.org/10.1016/j.neuron.2018.07.025, PMID: 30092215

Roux et al. 2022. Oscillations support short latency co-firing of neurons during human episodic memory formation. *eLife* 2022;11:e78109. DOI: https://doi.org/10.7554/*eLife*.78109 1 of 27

---

## [Author Response]

[Editors’ note: the authors resubmitted a revised version of the paper for consideration. What follows is the authors’ response to the first round of review.]

Summary:This is a very intriguing paper showing how hippocampal local field potentials couple with the activity of other cortical regions. This mechanism has been and continues to be extensively studied in other mammals, and thus its existence and relevance in humans is exciting. The reviewers were unanimous in their opinion that this work is worthy of publication in eLife, but pointed to key aspects of the paper that they felt were insufficient.Essential revisions:1. Hippocampal Data:A) Critically, the authors should include similar analyses of hippocampal neurons (if they exist) and explain why they don't have them otherwise.

In our initial submission, we reported that neurons outside the hippocampus, primarily in the MTL, phase-locked to hippocampal theta. However, we did not examine phase locking of hippocampal neurons, instead citing relevant findings in the literature. We agree that expanding our analyses to include phase locking of hippocampal neurons would enhance our contribution, allowing readers to draw conclusions about how phase-locking to hippocampal theta differs between hippocampal and extrahippocampal neurons. Inclusion of these new analyses also avoids asking readers to make comparisons across studies that used different methods. In an expanded dataset that now includes 391 hippocampal neurons in 27 subjects (Table 1), we describe several new results in hippocampal neurons, as detailed in Results lines 170-172 and 205-214 and Discussion lines 304-306 and 360-377. Our principal findings are:

Hippocampal neurons phase-lock to low frequency hippocampal oscillations at substantially higher percentages (60%, compared to 40% phase-locking for EC neurons and 30% for amygdala neurons) than we found for neurons in any extrahippocampal region (Figure 3A).Hippocampal neurons phase-lock to all three oscillations that are present in the hippocampal micro-LFP signal between 1-30Hz (see our response to Essential Revision 1B), corresponding to slow theta (2-4Hz), fast theta (6-10Hz), and β rhythms (13-20Hz) (Figure 2B,E,H and Figure 3B,C). In contrast, phase-locking to hippocampal LFPs by neurons outside the hippocampal is restricted the theta range, with most phase-locking occurring to slow hippocampal theta.

B) The nature of the theta oscillations needs to be explicated – are they short bursts accompanying eye movements or more sustained periods? – and example spectrograms and traces shown.

We agree that characterizing hippocampal theta oscillations in humans is important for interpreting our unit phase-locking results, and our manuscript benefits by addressing this issue directly. The suggestion to examine eye movement-related signals is also interesting, although we unfortunately did not collect eye-tracking data in any portion of our dataset. In response to the reviewers’ concerns, we added a new section of Results (lines 90-128) and corresponding figures (Figure 1 and Figure 1—figure supplement 1) that describe theta and other low frequency oscillations in the hippocampus using complementary approaches. First, following a suggestion by Reviewer 3, Figure 1 now includes several example hippocampal spectrograms and LFP traces from single subjects, along with accompanying description in Results lines 98-102:

“Many individual electrodes showed peaks in spectral power that rose above the background 1/f line in session-averaged LFP spectrograms (Figure 1A), indicating the potential presence of oscillatory activity (Donoghue et al., 2020). The frequency and magnitude of these spectral peaks varied considerably across subjects (compare Figure 1A subpanels) yet were nearly exclusively observed between 2-20Hz.”

We also implemented a well-validated method called Better OSCillation (BOSC) detection, which identifies oscillatory bouts in time-domain LFP traces as intervals with high narrowband LFP power above the background spectrogram that is sustained for a minimum duration to be considered oscillatory. The details of this procedure are described in Methods lines 455-469:

“For each LFP channel, we identified time-resolved oscillatory bouts at the 30 frequencies defined in the previous section using the BOSC (Better OSCillation) detection method, as described previously (Whitten et al., 2011). BOSC defines an oscillatory bout according to two threshold criteria: a power threshold, P_T_, and a duration threshold, D_T_. P_T_ is set to the 95th percentile of the theoretical Χ^2^ probability distribution of power values at each frequency, under the null hypothesis that powers can be modeled as a straight power law decaying function (the ‘1/f’ spectrum). Defining P_T_ for each frequency of interest requires first finding a best fit for 1/f. We obtained this fit by implementing the recently developed FOOOF (Fitting Oscillations & One-Over F) algorithm, which uses an iterative fitting procedure to decompose the power spectrogram into oscillatory components and a 1/f background fit (Donoghue et al., 2020). To avoid assuming that the 1/f spectrum was stationary across the recording session, we divided the LFP into 30s epochs and re-fit 1/f (and P_T_, by extension) in each epoch. Finally, we set D_T_ = 3/f, consistent with the convention used in previous studies (Ekstrom et al., 2005; Watrous et al., 2011; Aghajan et al., 2017), such that power at a given frequency f must exceed P_T_ for a minimum of 3 cycles for an oscillatory bout to be detected.”

In Figure 1C, we show that BOSC identifies hippocampal oscillations in three frequency bands corresponding to slow theta (2-4Hz), fast theta (6-10Hz), and β oscillations (13-20Hz). Slow theta was the most prevalent oscillation, detected ~6% of the time at the peak frequency of 3Hz, across subjects. These results are broadly consistent with findings from macroelectrode EEG signals in previous studies, and our results constitute the first analysis of hippocampal oscillation prevalence across frequencies in human microwire recordings. We believe that this addition to our manuscript will therefore be of general interest to scientists studying hippocampal physiology in humans, as we confirm broad consistency across greatly differing spatial scales.

C) The authors should describe the anatomical locations of the hippocampal LFP electrodes, explain how they were chosen, and whether they varied from subject to subject.

Electrode placement was determined solely by clinical teams, independent of the goals of our research program. After surgical implantation, we confirmed the location of each depth electrode tip using post-operative structural MRIs or post-operative CT scans coregistered to pre-operative MRIs, as described in Methods lines 407-408 and 420-427. While these depth electrode tips are visible on CT/MRI, it is unfortunately difficult to visualize the microwires that extend 5mm further, and from which our unit recording and LFP signals derive. Microwires are known to splay out unpredictably during implantation, creating a 5mm radius of uncertainty as to the location of any recorded unit or micro-LFP. For this reason, we did not attempt to resolve electrode locations at the level of hippocampal layers or subfields, instead electing to use coarser anatomical labels for which we had a higher degree of confidence. In the revision, we have added a Methods sentence (lines 427-430) that seeks to clarify this limitation. Additionally, while we believe that it would be interesting to examine differences in hippocampal theta properties along the longitudinal axis, a large majority of hippocampal electrodes in our dataset were implanted in the anterior hippocampus, just posterior to the amygdala, due to decisions by the surgical team made on clinical grounds. We unfortunately lacked sufficient sample size to analyze anterior versus posterior hippocampal differences. We have added this explanation as a limitation in the Discussion (lines 356-359).

2. New Analyses: The authors focus on linking hippocampal LFP and extra-hippocampal spikes. In addition to hippocampal LFP -> hippocampal spikes (point 1A), it would be valuable to assess hippocampal LFP -> extra-hippocampal LFP. This might include looking at dynamics of theta-band coherence and/or theta-modulation of high γ.

We agree that assessing relations between hippocampal and extrahippocampal LFPs would enhance the impact and aid interpretation of our findings. We addressed this suggestion by analyzing cortico-hippocampal LFP-LFP relations using a method other than coherence (as the reviewer suggests) that captures the same idea but is more consistent with our method for phase-locking quantification. Specifically, we calculated the Dice coefficient – a measure of overlap percentage, or co-occurrence – between oscillatory bouts in the hippocampus and each extrahippocampal region (see Methods lines 507-515). We find that hippocampal theta oscillations are often accompanied by theta bouts in the EC and (to a lesser extent) in the amygdala, both regions where neuron-LFP phase-locking to hippocampal theta is high. However, we also observe highly overlapping theta oscillations in the hippocampus and parahippocampal gyrus (PHG) despite the relative absence of PHG neuron phase-locking to hippocampal theta. These data suggest that oscillatory synchrony contributes to but does not fully explain the extrahippocampal neuron phase-locking to hippocampal theta phenomenon. The results are visualized in Figure 4A and described in Results lines 247-261.

3. Data Analysis Issues:A) While they explain that many of their spikes are coming from multi-units, they should be more upfront about this, and when possible, try to classify neurons as pyramidal or interneurons, and show results by neuron type / base firing rate.

We agree with the reviewers and have revised our paper to remove reference to “single-unit” or “single-neuron” activity; instead, we now refer to our analyses of “single- and multi-neuron recordings.” This distinction is clarified in the revised Abstract (line 19), Introduction (line 76), Results (lines 82-85), and Discussion (lines 299-301). While we refer to example “neurons” that phase-lock to oscillatory signals, we now clarify at the top of the Results (lines 82-85) that this term is used as a shorthand reference to single- as well as multi-unit firing patterns, which we do not distinguish between. We expanded the spike-sorting section of our Methods to explain, in greater detail, how we separated spikes from each microwire channel into one or multiple units (lines 433-442). Finally, we added three sentences of discussion explaining our decision not to attempt separating single- from multi-units or excitatory from inhibitory single-units, and how this might influence how our findings are interpreted (lines 381-390):

“A second limitation concerns the quality of unit isolation, as we recorded spikes from single microwires with limited ability to resolve spiking contributions from different neurons. Although some studies in humans have attempted to distinguish between single-units and multi-units and between excitatory and inhibitory neurons, unit quality metrics from microwires do not leave us with high confidence in the accuracy with which these distinctions can be made, while the potential for better quality unit recordings using tetrodes or Neuropixels may soon provide clarity with respect to cell-type-specific differences (Despouy et al., 2020; Chung et al., 2022). In the meantime, we believe it is unlikely that this limitation would change any of our main conclusions, which do not depend on knowing if a unit is truly “single” versus a combination of several neighboring cells.”

B) Critically, there is concern that some of the analyses use the same set of data to e.g., pick frequency bands and analyze coupling, rather than using some form of cross-validation.

We thank the reviewers for catching this error, which introduced bias in favor of finding the stated effect. Specifically, we had reported that neurons outside the hippocampus still phase-locked to hippocampal theta after we controlled for local theta phase-locking effects. However, Reviewer 2 noted that our method used the same data to select a

neuron-specific, peak hippocampal phase-locking frequency and then proceeded to analyze phase-locking strength between local and hippocampal LFPs at that frequency. Through statistical double-dipping, this method introduced bias toward finding greater phase-locking to hippocampal than local LFPs, all else being equal.

In our revision, we have corrected this statistical error. Our revised method leverages the fact that we now distinguish between oscillatory and non-oscillatory intervals by applying a combination of LFP power and duration thresholds to the time-domain LFP (see our reply to Essential Revision 1B). Moreover, we only analyze phase-locking to the hippocampus during ongoing oscillations. To reevaluate whether phase-locking to hippocampal oscillations occurs independently of local LFP phase-locking, we subdivide spikes into two groups: (1) spikes that occurred when local and hippocampal oscillations were both present, and (2) spikes that occurred when only hippocampal oscillations were present. To control for sample size differences between these groups, for each neuron we select the same number of spikes at each frequency. We then calculate phase-locking strength and significance within each group, using identical methods. We also show results from a control analysis in which phase-locking strength to local oscillations is analyzed based on the presence or absence of co-occurring hippocampal oscillations (Figure 4—figure supplement 2). We find that phase-locking to hippocampal theta is approximately twice as strong in the presence versus absence of local theta oscillations (Figure 4). However, significant numbers of EC and amygdala neurons still phase-lock to hippocampal theta when local theta is below the threshold for detection. This result is consistent with our original conclusion that extrahippocampal neuron phase-locking to hippocampal theta cannot be fully explained by local theta phase-locking and LFP-LFP theta synchrony. These analyses are described in Methods lines 516-532 and Results lines 262-297.

Reviewer #1:Hippocampal theta oscillations are among the most prominent rhythms in the mammalian brain. Extensive research in rodents has shown that neurons not only within the hippocampus but in widespread cortical areas can be phase-locked to hippocampal theta. Such cross-regional communication within theta frames has been postulated to be the foundation of many hippocampal operations. While previous studies in humans have documented the relationship between LFP theta and spiking in the hippocampus, coupling between hippocampal LFP and more remote cortical areas have not been demonstrated in human subjects. This is the topic of the present work. The authors show that spikes of single (and mostly multiunit) neurons in multiple cortical regions both in the same and opposite hemipheres are phase locked to transient occurrence of hippocampal theta LFP in the 2-6 Hz range. However, phase-locking is stronger in structure know to be part of the 'limbic system', such as the amygdala and entorhinal cortex. Theta phase locking was stronger to hippocampal than to local LFP and the magnitude of spike phase locking increased when the power of theta increased, associated with increased high frequency power. The results are straightforward and the analysis methods are reliable. The novel information is limited but informative and documents a missing aspect of theta communication in the human brain.

We thank the reviewer for their positive comments and helpful suggestions. We have made multiple changes to the manuscript to address their concerns.

Comments:1. Given the simple message, the text is a bit long with many repetitions and loose ends. This applies to both Introduction and Discussion. Potential implications to learning, etc are interesting but the findings do not provide additional clues, thus those aspects of the discussion are mainly distractions. Instead, perhaps the authors would like to discuss potential mechanisms of remote unit entrainment. They are talking about multi-synaptic pathways but these are unlikely to be a valid conduit. Instead, the septum, entorhinal cortex or retrosplenial cortex, with their widespread projections, may be responsible for coordinating both hippocampal and neocortical areas.

We agree with the reviewer and have removed the more speculative aspects of our Discussion, refocusing around topics that are most directly related to and informed by our results. We shortened both the Introduction and Discussion, decreased redundancies, and strove to increase the clarity of our exposition.

2. Arguably, the weakest part of the manuscript is the lack of hippocampal neurons. The authors refer to their own previous papers, but in a story which compares hippocampal theta oscillations with remote unit activity, it is strange that the magnitude of theta phase-locking to local hippocampal neurons is not available for comparison.

We agree that the lack of hippocampal neuron data is a limitation of our original study. In the resubmission, we include phase-locking comparisons between 391 hippocampal units from 27/28 subjects and locally-recorded microwire LFPs (Table 1). Identical methods are used to analyze hippocampal and extrahippocampal neuron phase-locking to permit direct comparison of the results. With respect to both the percentage of phaselocked neurons (~60%;) and mean phase-locking strength at the population level, we find that hippocampal neurons phase-lock more prominently to hippocampal oscillations than do neurons in any other region (Figure 3). By comparison, ~40% of EC neurons and ~30% of amygdala neurons phase-lock to the hippocampus, while minimal phaselocking is observed in more distal regions. Within the 1-30Hz frequency range that we analyze, hippocampal and extrahippocampal neurons both phase-lock predominately to slow theta (2-4Hz, all regions) or fast theta (6-10Hz, EC and hippocampal neurons) oscillations, while only neurons within the hippocampus phase-lock to higher-frequency β oscillations (13-20Hz) that are nested within the slow theta rhythm (Figure 2 and Figure 3B,C). Altogether, these results are consistent with our original findings that suggested a privileged role for hippocampal theta in coordinating spiking within the MTL, while also emphasizing that hippocampal theta entrainment is greatest for neurons within the hippocampus compared to neurons in densely-connected, neighboring regions.

3. How was the hippocampal LFP reference site chosen and did it vary substantially from subject to subject? Anterior or posterior locations?

The placement of electrodes was determined by the surgical team and was not related to our research goals. After surgery, we confirmed the location of each electrode using post-operative MRI or CT scans. It is difficult to visualize the microwires that extend from the electrodes and from which we recorded unit and LFP signals. These microwires can splay out unpredictably during implantation, leading to uncertainty in the location of the recorded units or micro-LFPs. Therefore, we did not attempt to resolve electrode locations at the level of hippocampal layers or subfields, instead using coarser anatomical labels. We have added Methods (lines 407-408 and 420-430) to clarify this limitation. Most hippocampal electrodes were implanted in the anterior hippocampus, with unfortunately not sufficient variability to permit analyzing differences along the longitudinal axis. We now note this limitation in the Discussion (lines 356-359).

4. The authors list 1233 single neurons but in the discussion they make it clear that most of them were multiple neurons. This should be emphasized up front and may be used as an excuse why the authors did not attempt to separate pyramidal cells from interneurons (interneurons have a much higher propensity to be entrained by projected rhythms).

We agree with the reviewer's concern about the lack of clarity regarding the nature of extracellular neuron recordings. In the revised manuscript, we have removed references to "single-unit" or "single-neuron" activity and instead refer to analyses in "single- and multi-neuron recordings" at several points in the paper where readers are likely to take note: in the revised Abstract line 19, Introduction line 76, Results lines 82-85, and Discussion lines 299-301. While we refer to example “neurons” that phase-lock to oscillatory signals, we now clarify at the top of the Results (lines 82-85) that this term is used as a shorthand reference to single- as well as multi-unit firing patterns, which we do not distinguish between. We expanded the spike-sorting section of our Methods to explain, in greater detail, how we separated spikes from each microwire channel into one or multiple units (lines 433-442). Finally, we added three sentences of discussion explaining our decision not to attempt separating single- from multi-units or excitatory from inhibitory single-units, and how this might influence how our findings are interpreted (lines 381-390):

“A second limitation concerns the quality of unit isolation, as we recorded spikes from single microwires with limited ability to resolve spiking contributions from different neurons. Although some studies in humans have attempted to distinguish between single-units and multi-units and between excitatory and inhibitory neurons, unit quality metrics from microwires do not leave us with high confidence in the accuracy with which these distinctions can be made, while the potential for better quality unit recordings using tetrodes or Neuropixels may soon provide clarity with respect to cell-type-specific differences (Despouy et al., 2020; Chung et al., 2022). In the meantime, we believe it is unlikely that this limitation would change any of our main conclusions, which do not depend on knowing if a unit is truly “single” versus a combination of several neighboring cells.”

5. Given that units were mixed, a logical extension would be to examine how hippocampal theta phase modulates high γ in neocortical areas. This could potentially yield a much larger data base, targeting the same question.

We agree with the reviewer that analyzing hippocampal theta phase modulation of cortical high γ power would be a very interesting extension of our current findings. However, in considering this suggestion together with other requested revisions, we decided that a convincing demonstration of inter-regional theta-γ coupling would require multiple additional analyses whose results would then need to be interpreted alongside our primary findings relating neuronal spikes to hippocampal theta phase. As our resubmission includes several new analyses that the reviewers deemed critical for clarifying our results, we respectfully decided that this analysis, while interesting, is not necessary to interpret our results at the unit level and may be best addressed in future work.

6. In the Discussion, the authors suggest that cross-regional theta phase coupling could be related to learning and other cognitive performance. However, spike-LFP coupling and coherence is confounded by LFP power increase and the authors cite Herweg et al., 2020 which did not find a relationship between theta power and memory performance. Is it then not logical to assume that cross-regional coupling may also not be related to memory?

We agree with the reviewer that our original submission introduced difficult-to-interpret confounds between hippocampal LFP power and phase, given that we analyzed all spikes from extrahippocampal neurons without regard to whether hippocampal LFPs had sufficient power to resolve phase at a given frequency. While our focus in the present submission is to describe novel physiological relations between hippocampal oscillations and unit firing, rather than further relating these characteristics to behavior, we have modified our phase-locking detection method to resolve the power/phase confound that the reviewer notes. Specifically, we now distinguish between oscillatory and non-oscillatory intervals by applying a combination of LFP power and duration thresholds determined using a well-established method (see Figure 1B,C and our reply to Essential Revision 1B). We proceed to analyze phase-locking to the hippocampus only during ongoing oscillations at a given frequency. This requirement enforces that any reported phase-locking coincides with times that hippocampal LFP power significantly exceeds the 1/f distribution for a sufficient duration to be deemed oscillatory, and our method further disentangles oscillatory bouts from asynchronous, high-amplitude events (sharp-wave ripples, interictal spikes, or movement-related artifacts) that are excluded from the analysis. Lastly, although we do not analyze behavior at present, our approach will be straightforward to extend to future studies relating phase-locking and spectral power as potentially separate predictors of memory performance.

7. Line 36. "Long-term potentiation and long-term depression in the rodent hippocampus are also theta phase-dependent (Hyman et al., 2003)." Pavlides et al. (Brain Res 1988) or Huerta and Lisman (Neuron 1995) are perhaps more relevant references here.

We thank the reviewer for this suggestion and have updated these references as recommended.

8. Line 82: "significant neocortical and contralateral phase-locking suggests". This is a strange phase. Perhaps significant phase locking of neurons in the neocortex in both hemispheres or similar would be a better formulation.

We agree with the reviewer and have removed this reference from the revised submission.

Reviewer #2:In this study, Schonhaut et al., describe the phase locking statistics of cortical and subcortical neurons with respect to hippocampal local field potential (LFP) recorded in 18 epilepsy patients undergoing seizure monitoring. Nearly 30% of extrahippocampal neurons showed phase locking to some bandpassed hippocampal signal. Amygdalar and entorhinal neurons were more likely to be phase locked, as compared to neurons recorded in other neocortical sites. Most neurons showed the strongest phase locking to hippocampal theta (2-8 Hz), though neocortical and amygdalar neurons tended to phase lock to lower theta bands. Spikes that were phase locked to hippocampal rhythms occurred during local LFP-states that showed moderate correlations with the spectral patterns observed in the hippocampus. These data are interpreted within the broader "communication through coherence" hypothesis.Large N, multi-region, single unit studies from humans are rare and the kind of mesoscopic descriptive analyses provided here serve as an important bridge between the large rodent literature on hippocampal physiology and human physiology and cognition. That said, there are some weaknesses in the analyses that could be addressed in a revised report. Also, a deeper discussion of the biological origin of human theta is merited in the discussion to address alternate explanations – beyond communication through coherence – of the data.

We thank the reviewer for their valuable comments to our manuscript. We have made multiple changes to the revised submission to address their concerns.

A similar statistical mistake was made several times. The author's logic goes like this: find the argmax in one sample, take the argument that generated that max, and use that to sample in another condition, and report that the max is higher in the first condition than the second. For example, on pg. 6 "This is difficult to reconcile with our results, in which 248/362 neurons (68.5%) phase-locked more strongly to hippocampal LFPs than to locally-recorded LFPs at their preferred hippocampal phase-locking frequency." The same flaw can be seen in Figure 5, where the spikes are sub-sampled to occur during strong phase locking in one condition, thus almost guaranteeing high power in the frequency bands that generated that strong phase locking (which was observed). This is a case in which cross-validating the data may be useful. The authors could take a subset of the hippocampal data to define the preferred frequency, and then test phase locking on the held out data from the hippocampus and cortex.

We thank the reviewer for identifying this limitation, which likely biased us toward finding the stated effect. In our original submission, we reported that neurons outside the hippocampus remained phase-locked to hippocampal theta after controlling for local theta phase-locking effects. However, as the reviewer notes, our method used the same data to select a neuron-specific peak hippocampal phase-locking frequency and then analyze phase-locking strength between local and hippocampal LFPs at that frequency, introducing an artifactual bias favoring greater phase-locking to hippocampal than local LFPs.

Our revision redesigns this analysis to correct the statistical error. Our new method leverages the fact that we now distinguish between oscillatory and non-oscillatory intervals by applying a combination of LFP power and duration thresholds to the timedomain LFP (see our replies to the next two comments for further details on this procedure). To evaluate whether phase-locking to the hippocampus occurs independently of phase-locking to local LFPs, we subdivide spikes into two groups: (1) spikes that occurred when local and hippocampal oscillations were both present, and (2) spikes that occurred when hippocampal oscillations were present in the absence of local oscillations. Phase-locking is then analyzed separately within each group, matching the number of spikes at each frequency, such that phase-locking strength to the hippocampus can be directly compared given the presence or absence of simultaneous local oscillations. We also show results from a control analysis in which phase-locking strength to local oscillations is analyzed based on the presence or absence of co-occurring hippocampal oscillations (Figure 4—figure supplement 2). Our full approach is described in Methods lines 518-532:

“For each extrahippocampal neuron, we analyzed phase-locking after dividing spikes into two groups according to the following criteria: (1) BOSC-detected oscillations were present in both the hippocampus and the neuron’s local region, or (2) BOSC-detected oscillations were present in the hippocampus but not the neuron’s local region (Figure 4). In a supplemental analysis Figure 4—figure supplement 2, we used the same approach but compared: (1) co-occurring local and hippocampal oscillations to (2) BOSC-detected oscillations that were present in a neuron’s local region but not the hippocampus. Phase-locking strengths were calculated at each frequency, in each spike group, and significance determined relative to null distributions as described in “Phase-locking strength and significance.” As chance-level phase-locking values are sample size dependent, for each neuron we matched the number of spikes in each group, at each frequency, excluding neurons with insufficient sample size (<50 spikes at any frequency). For example, for neuron i at frequency j, if 200 spikes occurred during co-present local and hippocampal oscillations and 150 spikes occurred when only hippocampal oscillations were present, we selected 150 spikes from the first group at random and proceeded to calculate phase-locking strength in each group.”

We find that significant numbers of EC and amygdala neurons still phase-lock to hippocampal theta in the absence of a local theta rhythm, although the number of phase-locked neurons is reduced compared to when local theta is present (Figure 4B). This result is consistent with our original conclusion that extrahippocampal neuron phase-locking to hippocampal theta cannot be fully explained by local phase-locking and LFP-LFP theta synchrony. These findings appear in lines 271-285:

“FDR-corrected phase-locking rates during co-occurring local and hippocampal oscillations were comparable to phase-locking rates when all spikes were included (see Figure 4A), with high phase-locking to hippocampal oscillations occurring among neurons in the EC and amygdala and minimal phase-locking among neurons in other regions. In contrast, when hippocampal oscillations occurred without co-occurring local oscillations, phase-locking rates to the hippocampus declined by nearly two-thirds in the EC (from 39% of neurons to 14%) and by half in the amygdala (from 28% to 15%), while phase-locking to the hippocampus in other regions mostly vanished. Phase-locking strength to the hippocampus decreased specifically at theta frequencies, and even neurons that remained significantly phase-locked in the absence of local oscillations showed reduced phase-locking strength (Figure 4C). We also considered the reverse analysis, asking whether phase-locking to local oscillations depended on the presence of co-occurring oscillations in the hippocampus. While local phaselocking rates in the amygdala and neocortex were unaffected by hippocampal oscillation presence, in the EC the percentage of locally phase-locked neurons was reduced by more than half when hippocampal oscillations were absent (Figure 4—figure supplement 2).”

The relationship between power and phase locking is not fully controlled in this paper. The phase seems to be calculated irrespective of whether there is any instantaneous power at that frequency band, introducing noise. This will bias away from finding significant phase locking to frequency bands that occur transiently. Therefore, I recommend defining some threshold of the existence of the spectral signal prior to using that signal to calculate phase.

We agree with the reviewer that the relation between LFP phase and power was not well controlled in our original submission, given our decision to analyze all spikes irrespective of whether they occurred during oscillatory states or not. Although this approach is consistent with how many previous studies have analyzed phase-locking in human intracranial EEG, it raises questions for our ability to interpret phase-locking to oscillatory events, given that hippocampal oscillations occur only sporadically in humans during virtual navigation and other stationary tasks.

In the revision, we address this concern by excluding spikes that occurred in the absence of ongoing oscillatory bouts. We employ a well-validated oscillation detection method that applies a combination of LFP power and duration thresholds to the timedomain EEG, as described in Methods lines 455-506 and Results lines 90-111. Please see our reply to the next comment for further details on this method.

A related point has to do with the nature of the theta rhythm in the human. There has been considerable controversy over the years as to whether this is a comparable signal to that studied in the rodent. Based off the citations in this manuscript, and the nomenclature of the spectral band, the authors seek to make explicit the commonality of the underlying physiology, or function. Rodent theta is a sustained rhythm, while primate theta seems to come in bouts, perhaps even related to sampling statistics, such as saccades, leading to the suggestion that the apparent theta may be better thought of as semi-rhythmic evoked responses. How long were the bouts of high theta power? Was eye movement tracked? If so it would be important to relate the signal to eye movements. If the low frequency signal is phase locked to eye movement and potentially reflects semi-rhythmic information arriving to (from?) the hippocampus, then a stronger case could be made that hidden "third parties" synchronize the apparent communication through coherence observed here, and in fact there may be no communication at all.

We thank the reviewer for their question concerning the nature of hippocampal theta oscillations in humans, which our original submission did not address. Unfortunately, we did not collect eye-tracking data in the present dataset. In the revised submission, however, we now include a new section of Results (lines 90-128) and corresponding figure describing hippocampal theta and other detected oscillations in the 1-30Hz range. We first show that time-averaged spectrograms from individual subjects commonly show narrowband peaks in the theta frequency range (Figure 1A). We then implement a well-validated oscillation-detection algorithm called BOSC (Better OSCillation) detection method, which identifies intervals in the time-domain EEG that LFP power at a given frequency significantly exceeds the background 1/f distribution for a minimum 3cycle duration (Figure 1B). The details of this procedure are described in Methods lines 455-475. In Figure 1C, we show that BOSC identifies hippocampal oscillations in three frequency bands corresponding to slow theta (2-4Hz), fast theta (6-10Hz), and β oscillations (13-20Hz). Slow theta was the most prevalent of these oscillations, detected ~6% of the time at the peak frequency of 3Hz, across subjects. These results are broadly consistent with findings from macroelectrode EEG signals in previous studies, and our results constitute the first cross-frequency analysis of hippocampal oscillation prevalence in human microwire recordings. We believe that this addition to our manuscript will be of general interest to scientists studying hippocampal physiology in humans, as we confirm broad consistency across greatly differing spatial scales.

The authors dedicate much of their discussion to relating the current result to the communication through coherence analyses. Oddly, LFP coherence was never addressed. A strong prediction of the current framing would be that: when coherence is high, phase locking should by high, and higher than other moments when power in either region is high but coherence is not observed. The authors should directly measure how phase locking is modulated by coherence.

We agree with the reviewer that it would be valuable to characterize the effect of LFP coherence on inter-regional phase-locking. We address this suggestion in the revision by first analyzing LFP-LFP relations using a method other than coherence (as the reviewer suggests) that captures the same idea but is more consistent with our method for phase-locking quantification. Specifically, we calculate the Dice coefficient – a measure of temporal overlap, or co-occurrence – between oscillatory bouts in the hippocampus and each extrahippocampal region (see Methods lines 507-515). We find that hippocampal theta oscillations are often accompanied by theta bouts in the EC and (to a lesser extent) in the amygdala, both regions where neuron-LFP phase-locking to hippocampal theta is high. However, we also observe highly overlapping theta oscillations in the hippocampus and PHG despite the relative absence of PHG neuron phase-locking to hippocampal theta. These data suggest that oscillatory synchrony contributes to but does not fully explain the extrahippocampal neuron phase-locking to hippocampal theta phenomenon. The results are visualized in Figure 4A and described in Results lines 247-261:

We also address the reviewer’s prediction that extrahippocampal neuron phase-locking to hippocampal theta is enhanced by local and hippocampal theta coupling. Retaining our definition of LFP-LFP relations as the temporal overlap between oscillatory bouts in each region, we compare neuronal phase-locking strengths at the population level given the presence or absence of co-occurring local oscillations. We find that phase-locking to hippocampal theta is approximately twice as strong when local theta present versus absent (Figure 4D), consistent with the reviewer’s hypothesis. This finding is described in Results lines 286-297:

“Finally, we confirmed these findings at the population level by computing the mean phase-locking strength across all neurons in each region, without regard to phase-locking significance, while still matching the number of spikes at each frequency between conditions in which local and hippocampal oscillations cooccurred versus only hippocampal oscillations occurred. As in Figure 3C, when local and hippocampal oscillations co-occurred, EC and amygdala neurons both phase-locked strongly to slow hippocampal theta, phase-locking to fast hippocampal theta was restricted to EC neurons, and other regions showed negligible phase-locking to hippocampal oscillations at any frequency (Figure 4D, left subpanel). When local theta was absent, the strength of EC and amygdala neuron phase-locking to hippocampal theta was reduced by half, while still remaining well above chance (Figure 4D, right subpanel). Collectively, these results provide direct evidence that inter-regional LFP-LFP theta coupling augments but is not strictly required for extrahippocampal neuron phase-locking to hippocampal theta.”

The authors also lump together biological entities that should have difference phase locking behaviors. The amygdala is not a monolithic region, does phase locking differ by nucleus? Also, do fast spiking inhibitory cells differ from excitatory cells? The authors should relate their phase locking measure to mean firing rate to show that it is insensitive to lower level cell statistics. This is important since the conclusions of the study would be quite different if neurons in the entorhinal cortex had high rates which artifactually drove up phase locking values.

We appreciate the reviewer’s point about subregions of some of our recording regions, including the amygdala, being functionally and structurally heterogenous, which could underscore differing phase-locking characteristics. Unfortunately, we are unable to resolve neuron locations at sufficient resolution to answer this question analytically given limited ability to record from the same region across subjects (electrode locations are determined by clinical teams, strictly based on medical determination) and uncertainty regarding the exact location of microwire electrode tips. We have added text explaining these limitations more clearly in Methods lines 424-430:

“Microwires were packaged in bundles of eight high-impedance recording wires and one low-impedance wire that served as the recording reference. Each microwire bundle was threaded through the center of a depth probe and extended 5mm from the implanted end. As microwires splay out during implantation and cannot reliably be visualized on post-operative scans, electrode localizations are regarded with a ~5mm radius of uncertainty that preclude analyses at the level of regional substructures or hippocampal layers or subfields.”

A similar limitation concerns the quality of spike waveforms from single-microwire recordings and the confidence with which we can separate single- from multi-units and excitatory from inhibitory single-units. Although some studies in humans have attempted to make these distinctions, we prefer to interpret our data through the lens of single- to multi-unit associations for which we have higher confidence in the conclusions drawn. In the revision, we have removed references to "single-unit" or "single-neuron" activity and instead refer to analyses in "single- and multi-neuron recordings" at several points in the paper where readers are likely to take note: in the revised Abstract line 19, Introduction line 76, Results lines 82-85, and Discussion lines 299-301. While we refer to example “neurons” that phase-lock to oscillatory signals, we now clarify at the top of the Results (lines 82-85) that this term is used as a shorthand reference to single- as well as multiunit firing patterns, which we do not distinguish between. We expanded the spikesorting section of our Methods to explain, in greater detail, how we separated spikes from each microwire channel into one or multiple units (lines 433-442). Finally, we added three sentences of discussion explaining our decision not to attempt separating single- from multi-units or excitatory from inhibitory single-units, and how this might influence how our findings are interpreted (lines 381-390):

“A second limitation concerns the quality of unit isolation, as we recorded spikes from single microwires with limited ability to resolve spiking contributions from different neurons. Although some studies in humans have attempted to distinguish between single-units and multi-units and between excitatory and inhibitory neurons, unit quality metrics from microwires do not leave us with high confidence in the accuracy with which these distinctions can be made, while the potential for better quality unit recordings using tetrodes or Neuropixels may soon provide clarity with respect to cell-type-specific differences (Despouy et al., 2020; Chung et al., 2022). In the meantime, we believe it is unlikely that this limitation would change any of our main conclusions, which do not depend on knowing if a unit is truly “single” versus a combination of several neighboring cells.”

Reviewer #3:The manuscript by Schonhaut et al. presents novel analysis on an impressive dataset of more than 1200 neurons across diverse brain areas in the human brain to investigate their modulation by hippocampal theta oscillations. They found a substantive proportion of cells phase-locked to hippocampal activity, mainly in the theta frequency band, in several areas know to be functionally related to the hippocampus, some of them receiving monosynaptic hippocampal inputs but other only indirect ones. These results extend previous reports in humans showing hippocampal interactions with these structures but at the level of mesoscopic activity and highlight the ubiquity of spike-theta timing and the importance of single-unit studies in humans. Additional analysis, detailed below, will contribute to give a better description of the data, provide stronger support for some of the authors' claims and clarify some issues.

We thank the reviewer for their positive feedback and helpful comments to our manuscript.

1. I assume that the dataset also includes hippocampal units, why then excluding them from the analysis? Although the main novelty is in the coupling of cells in other structures with hippocampal LFPs, it would be useful to also compare it with the coupling of local hippocampal cells.

We thank the reviewer for this suggestion and agree that our study would benefit from including hippocampal units. In our revised submission, we include data from 391 hippocampal neurons in 27/28 subjects. We perform identical phase-locking analyses in these neurons as in neurons outside the hippocampus to permit direct comparison between them. With respect to both the percentage of phase-locked neurons (~60%;) and mean phase-locking strength at the population level, we find that hippocampal neurons phase-lock more prominently to hippocampal oscillations than do neurons in other regions (Figure 3). By comparison, ~40% of EC neurons and ~30% of amygdala neurons phase-lock to the hippocampus, while minimal phase-locking is observed in more distal regions. Within the 1-30Hz frequency range that we analyze, hippocampal and extrahippocampal neurons both phase-lock predominately to slow theta (2-4Hz, all regions) or fast theta (6-10Hz, EC and hippocampal neurons) oscillations, while only neurons within the hippocampus phase-lock to higher-frequency β oscillations (1320Hz) that are nested within the slow theta rhythm (Figure 2 and Figure 3B,C). Altogether, these results are consistent with our original findings that suggested a privileged role for hippocampal theta in coordinating spiking within the MTL, while also emphasizing that hippocampal theta entrainment is greatest for neurons within the hippocampus compared to neurons in densely-connected, neighboring regions.

Our findings show that hippocampal neurons have significantly higher percentages of phase-locking (~60%) to low-frequency hippocampal oscillations than neurons in any other region outside the hippocampus (Figure 3A). Furthermore, we show that hippocampal neurons phase-lock to all three frequency bands for which hippocampal oscillations above the background noise distribution were reliably observed (see Figure 1C): slow theta (2-4Hz), fast theta (6-10Hz), and β (13-20Hz) (Figure 2B,E,H and Figure 3B,C). This finding is in contrast to our phase-locking results in extrahippocampal neurons, which primarily phase-locked to slow hippocampal theta, although some entorhinal cortex neurons also phase-locked to fast hippocampal theta (Figure 2C,D,F,G,I,J and Figure 3B,C). Our phase-locking results in hippocampal neurons are now described in the revised Methods, Results, and Discussion sections, and are shown in Table 1, Figure 2, and Figure 3.

2. Include average power spectrum of hippocampal LFPs. Additional examples of raw LFP traces overlaid to spectrograms (perhaps in Supplementary) will help to illustrate the nature of hippocampal oscillations.

We thank the reviewer for this suggestion and have included these examples of timeaveraged LFP spectrograms and raw LFP traces in Figure 1A,B and Figure 2A.

3. The authors compared fractions of significantly modulated units and their preferred frequencies across regions. While very informative, these analyses are not sufficient to capture the richness of spike-LFP interactions likely existing in the dataset. Were there differences in the strength of phase-locking across regions? (this analysis could be added to Figure 2). Studies in rodents have shown that theta phase-locked units in different structures have characteristics preferred firing phases (when hippocampal LFP is used as a reference). Authors can easily look if this is also the case in their data. They should include both pooled data statistics of mean phases across regions and single neuron examples of firing probability by LFP phase (such examples could be added to the single unit plots in Figure 1).

We thank the reviewer for this suggestion and have added analyses describing differences in phase-locking strength at the population level, combining data across all neurons in each region. These results are now shown in Figure 3C and Figure 4D and confirm that EC and amygdala neurons phase-lock to hippocampal theta both at higher rates and greater magnitudes (i.e. phase-locking strength) than do neurons in more distal regions where phase-locking has been reported in rodents, notably mPFC. We also assessed statistical differences in phase-locking strength between regions, as summarized in Results lines 223-236:

“We confirmed these conclusions in a secondary analysis that examined the mean phase-locking strength at each frequency across all neurons in each region, regardless of individual phase-locking significance (Figure 3C). This approach benefited from not requiring an explicit significance threshold to be defined. Instead, we assumed that if the neurons in a given region did not phaselock measurably to the hippocampus, then the mean phase-locking strength across these neurons would approach zero (no difference versus the null distribution) at increasing sample size. Indeed, population phase-locking strengths were close to zero across frequencies for neurons in the PHG, STG, OFC, and ACC, consistent with the relative absence of individually phase-locked neurons in these regions. In contrast, neurons in the EC and amygdala both showed strong phase-locking to slow hippocampal theta frequencies, while EC but not amygdala neurons exhibited a secondary rise in phase-locking strength to fast hippocampal theta. Finally, neurons in the hippocampus showed stronger phase-locking to hippocampal oscillations at all frequencies than neurons in any other region, with peaks in phase-locking strength at all three (slow theta, fast theta, and β) oscillatory bands.”

As the reviewer suggests, we have added polar plots showing the distribution of spikephases for individual neuron examples in Figure 2. However, information about preferred firing phase across a population of neurons is unfortunately less revealing in humans than in animal models for several reasons related to currently available recording equipment of challenges of the clinical setting in which these recordings are conducted. First, there is a ~5 radius of uncertainty as to microwire electrode locations that acts as a spatial resolution limit for both unit and micro-LFP locations. Second, there is variability in recording locations from the same brain region across subjects due to implantation decisions made by the clinical teams, which complicates pooling data across subjects beyond a certain level of spatial coarseness. Third, the quality of singlemicrowire recordings in humans provides limited ability to resolve single- from multi-unit waveforms with high certainty for a large majority of units, which precludes making the more-difficult determination of single-unit subtypes. Although some studies in humans have attempted to make these distinctions, we prefer to interpret our data through the lens of single- to multi-unit associations for which we have higher confidence in the conclusions drawn.

In the revision, we attempt to be more clear about these limitations in several ways. First, we removed references to "single-unit" or "single-neuron" activity and instead refer to analyses in "single- and multi-neuron recordings" at several points in the paper where readers are likely to take note: in the revised Abstract line 19, Introduction line 76, Results lines 82-85, and Discussion lines 299-301. While we refer to example “neurons” that phase-lock to oscillatory signals, we now clarify at the top of the Results (lines 8285) that this term is used as a shorthand reference to single- as well as multi-unit firing patterns, which we do not distinguish between. We expanded the spike-sorting section of our Methods to explain, in greater detail, how we separated spikes from each microwire channel into one or multiple units (lines 433-442). Finally, we added three sentences of discussion explaining our decision not to attempt separating single- from multi-units or excitatory from inhibitory single-units, and how this might influence how our findings are interpreted (lines 381-390):

“A second limitation concerns the quality of unit isolation, as we recorded spikes from single microwires with limited ability to resolve spiking contributions from different neurons. Although some studies in humans have attempted to distinguish between single-units and multi-units and between excitatory and inhibitory neurons, unit quality metrics from microwires do not leave us with high confidence in the accuracy with which these distinctions can be made, while the potential for better quality unit recordings using tetrodes or Neuropixels may soon provide clarity with respect to cell-type-specific differences (Despouy et al., 2020; Chung et al., 2022). In the meantime, we believe it is unlikely that this limitation would change any of our main conclusions, which do not depend on knowing if a unit is truly “single” versus a combination of several neighboring cells.”

4. Did phase-locked and non phase-locked units have different properties? The authors can compare if they differ in basic properties such us mean firing rate, waveform width, inter-spike intervals, burstiness, etc., as it has been reported in other studies in non-human primates and rodents. These analyses could be extended to show if units with different properties also differ in their preferred phase-locking frequency, or phase. It would be very interesting if these analyses reveal the existence of heterogeneous cellular populations with different relation to hippocampal theta, even if the single-unit isolation quality is limited due to the low density recordings. In relation to this, authors should also plot unit auto-correlograms. ACGs can be computed for all the spikes, but also only for the strongly phase-locked spikes, to show if, at least during periods of strong oscillatory activity, some units show rhythmicity.

We thank the reviewer for this suggestion. We agree that exploring heterogeneity in neuronal populations through the lens of hippocampal theta phase-locking (and other physiological phenomena) is ultimately critical for connecting studies in humans with the wealth of knowledge from animal models, and for linking LFP/behavior associations in humans to richer accounts of neural circuit function. However, given the data quality limitations outlined in the previous response, and the possibility of performing higherquality, higher-density unit recordings from human subjects on the horizon, we are hesitant to perform analyses aimed at characterizing cellular heterogeneity at present given the likelihood of obtaining either uninterpretable null results or results due to noise/variability in recording conditions that we might wrongly interpret as signal. Our revisions outlined in the previous comment are aimed at conveying some of this thought process to the reader. We remain confident in the accuracy and value of our present findings to the subfield of human unit recordings, and hope that the reviewer may understand our hesitance toward some of these nonetheless valued suggestions.

5. To better interpret the results in Figure 4, it would be important to know if the recording sites in both hippocampi were from the same sub-region and similar location along the longitudinal hippocampal axis in each subject and if the degree of synchrony between the LFP in both hemispheres. Coherence or phase-locking between LFPs across hemispheres should be computed and also power spectrum for both of them shown.

Unfortunately, uncertainly regarding the exact location of recorded units and micro-LFPs makes it difficult to resolve information at the level of hippocampal subregions using presently available recording techniques in human subjects. We have added an explanation to the revised Methods to clarify this limitation, lines 446-454:

“Electrode localizations were confirmed by the clinical team using post-operative structural MRIs or post-operative CT scans co-registered to pre-operative structural MRIs. Microwires were packaged in bundles of eight high-impedance recording wires and one low-impedance wire that served as the recording reference. Each microwire bundle was threaded through the center of a depth probe and extended 5mm from the implanted end. As microwires splay out during implantation and cannot reliably be visualized on post-operative scans, electrode localizations are regarded with a ~5mm radius of uncertainty that preclude analyses at the level of regional substructures or hippocampal layers or subfields.”

Additionally, while we believe it would be interesting to examine differences in hippocampal theta along the longitudinal axis, a large majority of hippocampal electrodes in our dataset were implanted in the anterior hippocampus, just posterior to the amygdala, due to decisions by the surgical team on clinical grounds. We unfortunately lacked sufficient sample size to analyze anterior versus posterior hippocampal differences. We have added this explanation as a limitation in our revised Discussion (lines 374-376).

6. In Figure 4C-D it seems that phase-locking strength across hemispheres was not correlated but preferred frequency was. This should be quantified and mentioned in text before moving to the correlation in Figure 4E.

We thank the reviewer for this comment and agree that our initial analysis of contralateral phase-locking was insufficient. After reviewing feedback from the Editor and other reviewers, we decided to remove this analysis describing occasional, lowlevel phase-locking to hippocampal theta in the contralateral hemisphere. We agree with the reviewer that our original characterization of this phenomenon lacked depth and was unnecessary to support our main conclusion, that neurons in MTL regions structurally connected to the hippocampus phase-lock readily and at high rates to hippocampal theta. By removing this auxiliary analysis and refocusing on our primary results describing ipsilateral neuron-to-LFP associations, we were able to keep the revised manuscript at similar length as our original submission despite adding several analyses that the Editor and reviewers considered of high importance (characterizing hippocampal theta oscillations, adding analyses in hippocampal neurons, describing inter-regional theta associations, and resolving statistical concerns).

7. The analysis in Figure 5D should be complemented by also checking the LFP-LFP phase-locking between the local region and the hippocampus. Were periods of high LFP power correlation also reflect enhanced phase-phase coupling? Were the structures also more phase-synchronous during periods of stronger spike-LFP coupling? These analyses could provide a more direct support for the interpretation of the authors in line with the CTC hypothesis.

In response to Reviewer 1 and 2’s concerns over confounds between LFP power and phase in the original submission, we removed our analysis of phase-locking relations to inter-regional LFP power correlation in the revision. To address the reviewer’s question, however, we added an analysis comparing extrahippocampal neuron phase-locking to hippocampal theta and inter-regional LFP-LFP theta relations , using two complementary approaches. First, we show in Figure 4A that while overlap between theta oscillations in the hippocampus and extrahippocampal regions is similar to phaselocking patterns found at the unit level (high coupling between hippocampal theta and EC theta, moderate coupling to amygdala theta, and low coupling to mPFC theta), a dissociation is found in the strong coupling between hippocampal and PHG theta despite the relative absence of PHG neuron phase-locking. Therefore, LFP-LFP oscillation coupling and unit-LFP phase-locking are not interchangeable. Second, we directly compared extrahippocampal neuron phase-locking to hippocampal theta given the presence or absence of a detectable local theta rhythm. We found that phaselocking strength to the hippocampus was approximately doubled when hippocampal and local theta oscillations co-occurred versus when local theta was absent (Figure 4B-D). However, substantial numbers of EC and amygdala neurons still phase-locked to hippocampal theta even when local theta was below the detection threshold, echoing observations in rodents (Siapas et al., 2005) that local theta is not strictly required for hippocampal theta to entrain remote neuron populations.

8. Was there any relation of the "strongly phase-locked" periods with global variables reflecting brain state (e.g. drowsiness versus attention to the task, etc.) or with the firing dynamics of the units (instantaneous firing rate or inter-spike intervals)?

Unfortunately, the present data does not include any periods of sleep or other largescale changes in physiological state that would allow answering the reviewer’s question. We agree that this would be a very interesting direction for future research exploring brain-state and behavioral associations with our physiologically-grounded observation of inter-regional phase-locking to hippocampal theta. In the revision we include a comment on this point in Discussion lines 391-403:

“Still little is known about the relations between theta phase-locking and human cognition. Prior studies have focused on the behavioral correlates of phaselocking to local theta rhythms within the MTL; for example, successful image encoding was found to depend on theta phase-locking strength among hippocampal and amygdala neurons (Rutishauser et al., 2010), while another study found that MTL neurons can represent contextual information in their theta firing phase (Watrous et al., 2018). Here we show that hippocampal theta oscillations also inform the timing of neuronal firing in regions beyond the hippocampus, positioning theta oscillations at the interplay between local circuit computations and inter-regional communication. It remains unknown if behavioral or brain-state dissociations can be found between local and inter-regional phaselocking, or between spike-LFP and LFP-LFP phase synchronization. Such analyses could be well positioned to unite findings from animal and human studies and advance a more mechanistic account of hippocampal-dependent processes across multiple levels of scale, from single neurons to macroscopic fields.”

[Editors’ note: what follows is the authors’ response to the second round of review.]

The manuscript has been improved but there are some remaining issues that need to be addressed, as outlined below:Essential revisions:Reviewer #3 (Recommendations for the authors):1. At several parts in the manuscript the authors suggest that the coupling between extrahippocampal units to hippocampal oscillations is in the direction of LFP -> Spike (lines 168-169; 182-184; 238-239). I was surprised by this, because classically distal spike-LFP coupling is usually interpreted as directional coupling in the opposite direction (Spike -> LFP). This is because the common assumption would be that a if a spike in region A is coupled to the LFP in region B, then this is because the neuron in region A elicits post-synaptic currents in region B, hence spike -> LFP (see Buzsáki and Schomburg, 2015; Liebe et al., 2012; Jacob et al., 2018; see also Roux et al., 2022 for an example in the human MTL). Now, this doesn't mean that coupling in the other direction LFP ◊ Spike is not also possible, but it would require additional steps, whereby the LFP in region A entrains local neurons which then project to region B, where they elicit an LFP in region B which then induces firing of local neurons at specific phases. However, such an explanation is less parsimonious and thus requires additional evidence. Specifically, two additional analyses would be useful in this regard. First, the LFPs in region A and region B would have to be phase coupled. The authors demonstrate a coupling in terms of power, by measuring co-occurrence of 'bouts', but they do not demonstrate whether there is a consistent phase-relationship between the two regions where 'bouts' co-occur. Second, coupling measures which allow inferences about directionality should be applied to the spike and LFP time series. This can be done, for instance, by convolving the spike time series with a gaussian envelope and then applying the phase slope index to both time series (as done in Roux et al., 2022).

We found the reviewer’s motivation and recommendations for further analyses deeply thought-provoking, and look forward to further characterizations of phase-locking of single neurons to LFPs based on notions of directionality. However, here it was not our intention to imply a mechanism or direction of coupling, and we felt that would be best suited for subsequent investigations that stand on the strength of our initial description of this phenomenon. On the other hand, we appreciate why the reviewer was inclined to interpret our rhetoric in this way, and to render it consistent with our original intentions throughout the manuscript, we have removed all language suggestive of a particular directionality of spike–LFP coupling. In particular, we replaced mention of entrainment of single-neuron activity by the hippocampus with neutral references to phase-locking between spikes and LFPs. These changes address the reviewer’s justified concerns about speculation as to the mechanism or direction of the phase coupling phenomena described in the paper.

“This last step allowed us to directly compare phase-locking rates to local versus remote hippocampal influences. → This last step allowed us to directly compare phase-locking rates to local versus remote hippocampal oscillations.” (lines 188– 189)

“Phase-locking to local oscillations was nonetheless prevalent in the PHG (24%) and STG (49%), indicating that many of these neurons were rhythmically entrained, just not by oscillations in the hippocampus. → Phase-locking to local oscillations was nonetheless prevalent in the PHG (24%) and STG (49%), indicating that many of these neurons fired at specific phases of LFP oscillations, just not those recorded in the hippocampus.” (lines 202–204)

“Altogether, these results highlight a triad of regions in the hippocampus, EC, and amygdala with strong spike-time synchronization to hippocampal oscillations, while neurons in more remote, cortical regions known to interact with hippocampus-dependent processes (Eichenbaum, 2000; Squire, 2011; Ranganath and Ritchey, 2012) were minimally entrained by hippocampal rhythms. → Altogether, these results highlight a triad of regions — the hippocampus, EC, and amygdala — that features strong spike-time synchronization to hippocampal oscillations, while neurons in more remote, cortical regions that are known to interact with hippocampus-dependent processes (Eichenbaum, 2000; Squire, 2011; Ranganath and Ritchey, 2012) phase-locked minimally to hippocampal rhythms.” (lines 208–212)

“In the amygdala and remaining cortical regions, this balance shifted: only a few neurons phase-locked to fast hippocampal theta, while most neurons were entrained exclusively by slow theta. → In the amygdala and remaining cortical regions, this balance shifted: only a few neurons phase-locked to fast hippocampal theta, while most neurons coupled exclusively to slow theta.” (lines 239–241)

“Our data reveal that neurons not only within the hippocampus, but in remote regions — particularly the entorhinal cortex and amygdala — are entrained by hippocampal theta phase. → Our data reveal that neurons not only within the hippocampus, but in remote regions — particularly the entorhinal cortex and amygdala — phase-lock to hippocampal theta oscillations.” (lines 259–260)

2. Oscillatory bouts appear in three frequencies, ~3 Hz, ~7 Hz, ~ 15 Hz, with decreasing density of occurrence (i.e. 3 > 7 > 15 Hz). This could be indicative of an asymmetric 3 Hz oscillation which induces spurious signals at the first and second harmonic. Indeed, the 15 Hz oscillation appears to be coupled to the 3 Hz oscillation which would be consistent with this assumption (although the fact that 7 Hz is not, would not support this argument, but still). To counter this the authors should show how often bouts in the three frequency windows co-occur. If the higher frequencies are a reflection of asymmetric wave shapes then there should be a tight correlation with respect to when the bouts occur. Furthermore, the authors could investigate the waveshape of the 3 Hz oscillation by calculating an asymmetry index (as done in Roux et al., 2022; see Figure 5, figure supplement 1).

The reviewer raises an important question regarding the degree to which the three hippocampal LFP oscillations that we observed at 3, 7, and 15Hz peaks (Figure 1C) occurred independently. We followed the approach suggested by the reviewer in evaluating dependencies among these oscillations.

The Reviewer recommended that we examine the asymmetry of the average waveform for each of the 3, 7, and 15Hz hippocampal oscillations. To this end, we first visualized the mean waveform of the first three cycles of the oscillatory bouts at each of these frequencies, averaging first within and then across subjects. The figure below displays these waveforms (panel A). Although none of these waveforms exhibit asymmetry on inspection, we also sought to identify any asymmetries quantitatively. Following the method of Roux et al., 2022, as suggested by the Reviewer, we computed the asymmetry index during oscillatory bouts on each hippocampal LFP recording, at 3, 7, and 15Hz. Panel B shows the asymmetry index values (M ± SEM across participants) at each of these frequencies.

For each hippocampal recording included in our primary analyses (Figure 1), we first calculated the Dice similarity coefficient between oscillatory bouts at each peak oscillation frequency (3, 7, and 15 Hz) and all remaining frequencies from 1-30Hz, respectively. The Dice coefficient ranges from 0 to 1 and quantifies the degree of overlap between two binary vectors. In our analysis, these vectors comprised masks of oscillatory bout occurrence (see Methods section "Oscillatory bout identification") at two frequencies over time, within the same hippocampal electrode.

The resulting figure (see below) shows the Dice coefficients (M ± SEM across participants) in panel C. These plots reveal the overlap between the 3, 7, and 15Hz hippocampal oscillations and all other examined frequencies. If the 7 or 15Hz oscillations were attributable to spurious harmonic interactions with a lower frequency oscillation, we would expect these plots to show a multi-modal correlation structure. However, as can be seen in the figure, Dice coefficient relationships were unimodal for all three oscillations. We do not observe any clear correlations between oscillations at non-adjacent frequencies of a magnitude that would support concluding our results might be partially driven by spurious harmonics.

As we believe these analyses will also be of interest to our readers, we have included these results as a new supplementary figure (Figure 1—figure supplement 1).

We have also added the following description to the Methods (lines 504–519):

“All analyses to analyze waveform asymmetry were confined to oscillatory bouts as identified in the previous section. An inspection of the 3, 7, and 15Hz oscillations averaged during the time window corresponding to the first three cycles of each bout —1000 ms, 428 ms, and 200 ms, respectively — qualitatively assessed asymmetries in these waveforms. Then, an asymmetry index computed in keeping with previously established methods (Roux et al., 2022) quantified asymmetry in the 3Hz, 7Hz, and 15Hz waveforms. After initial preprocessing of the microwire LFPs (see “LFP preprocessing and spectral feature extraction'”), we applied a bandpass linear-phase Hamming-windowed FIR filter within a window of ±2Hz centered at the frequency of interest, and identified local maxima and minima in windows equivalent to a half-cycle at the frequency. After aligning these extrema in the filtered LFP trace to the nearest peaks and troughs within a quarter-cycle in the raw, unfiltered LFP trace, we found the average difference between the time taken to ascend from a trough to the next peak and to descend from the peak to the subsequent trough. We normalized this average difference to the range (-1, 1) by dividing by the cycle length f_s_/f, where f_s_ is the sampling frequency, and f is the frequency of interest, giving the asymmetry index value. The asymmetry index values for each hippocampal recording were averaged within subjects and then across subjects.”

Finally, we included a description of our findings from these analyses, as detailed above, to the Results (lines 123–139).

“Given that the prevalence of oscillatory bouts peaked at 3Hz, 7Hz, and 15Hz, we sought to verify that these frequency-wise clusters of oscillatory bouts were independent. For instance, an asymmetrical 3Hz rhythm in the hippocampus, analogous to the sawtooth-shaped theta waveform that rabbit, mouse, and rat hippocampus exhibits (Voytek et al., 2017), could generate harmonics at higher frequencies, inducing oscillatory bouts at the 7Hz and 15Hz components coincident with the 3Hz oscillatory bouts. However, visualizations of the average waveform of the first three cycles of the hippocampal oscillatory bouts indicate a symmetrical, sinusoidal oscillation at 3Hz, 7Hz, and 15Hz (Figure 1—figure supplement 1A). Computing an asymmetry index on the waveform at each of these frequencies throughout the recording session confirmed this result, yielding no significant asymmetry at 3 or 7Hz and a statistically significant but very small asymmetry at 15Hz, corresponding to an ascending flank less than a millisecond longer than the descending flank of the oscillation (Figure 1—figure supplement 1B). Finally, if the prevalence of oscillatory bouts at 7 and 15Hz arose from the harmonics of an asymmetrical 3Hz waveform, the oscillatory bouts should tend to occur at the same time, but examining the overlap of oscillatory bouts at 3Hz, 7Hz, and 15Hz with bouts at all other frequencies revealed no evidence for such a pattern of inordinate co-occurrence (Figure 1—figure supplement 1C). Therefore, hippocampal oscillatory bouts occur in three independent bands centered at 3Hz, 7Hz, and 15Hz.”

3. A recent paper that seems highly relevant to the current one is not mentioned. I am referring to the paper by Roux et al. 2022 who also investigated local and distal (cross-regional) spike-LFP coupling in the human MTL during a memory task. In line with the current study, Roux et al. also show distal coupling of MTL neurons in theta. Furthermore, they demonstrate that this coupling is related to memory whereby coupling at faster frequencies predicts successful formation of associations, whereas coupling to slow frequencies predicts unsuccessful formation of associations. Crucially, and central to the current study, Roux et al. 2022 demonstrate that coupling at theta frequencies was correlated with the latency of co-firing of pairs of distally coupled neurons. Therefore, several statements in the paper should to be revisited in light of that previous study (i.e. lines 398-400; 311-316; 66-75). I assume the reason for why the authors did not include that study is that it appeared late in 2022, likely when this paper was already submitted or close to submission. I also think that the Roux et al. paper does not take away anything in terms of novelty from this paper, as Roux et al. were not able to split up the distal coupling into the different MTL subregions due to a lower yield in neurons. If anything, the two papers nicely complement each other and both support a central role of MTL theta oscillations in routing information in the human brain in the service of memory and navigation.

We thank the reviewer for bringing to our attention the important work of Roux and colleagues in characterizing the phase-locking behavior of MTL neurons, and in particular, to LFPs in distal regions of the MTL. Given our own interest in the role of these phenomena in memory and cognition, we were pleased to see their intriguing first step in directly relating the phenomenon of phase-locking between MTL neurons and distal theta to human memory. We earnestly concur with the reviewer’s assessment that Roux et al., 2022 and our own paper reinforce the common theme that theta oscillations crucially support communication across regions in MTL, each pursuing this line of investigation in novel directions. Accordingly, in discussion of the existing literature and the relative contributions of our paper to the field, we have consistently acknowledged Roux et al., 2022, and at the points indicated by the reviewer in particular.

“However, few studies have investigated oscillatory phase coding of neuronal responses outside the hippocampus in humans. A recent study associated episodic memory with increased coupling between spikes in extrahippocampal MTL regions and distal theta with episodic memory, supporting the hypothesis that hippocampal theta facilitates interregional communication, especially with respect to memory and navigation (Roux et al., 2022). Nevertheless, that study did not distinguish the contributions of extrahippocampal and hippocampal oscillatory bouts, nor characterize the differential roles of MTL regions in this novel spike–phase coupling phenomenon.” (lines 72–78)

“Finally, although a prior study has examined distal spike LFP phase coupling (Roux et al., 2022), we provide the first direct evidence in humans that LFP–LFP coupling enhances spike-time synchronization between regions, as extrahippocampal neurons phase-lock approximately twice as strongly to hippocampal theta when local theta oscillations co-occur, as when local theta is absent.” (lines 331–335)

“Here we show that hippocampal theta oscillations also inform the timing of neuronal firing in regions beyond the hippocampus, positioning theta oscillations at the interplay between local circuit computations and interregional communication. In light of these results, and recent findings that the phaselocking characteristics of MTL neurons to local γ and distal theta, but not local theta and distal γ, distinguished successful memory (Roux et al., 2022), behavioral or brain-state dissociations between local and interregional phase-locking, and between spike–LFP and LFP–LFP phase-synchronization, merit further investigation.” (lines 415–421)